# Recombinant Adeno-Associated Viral Vectors (rAAV)-Vector Elements in Ocular Gene Therapy Clinical Trials and Transgene Expression and Bioactivity Assays

**DOI:** 10.3390/ijms21124197

**Published:** 2020-06-12

**Authors:** Thilo M. Buck, Jan Wijnholds

**Affiliations:** 1Department of Ophthalmology, Leiden University Medical Center (LUMC), 2333 ZC Leiden, The Netherlands; t.m.buck@lumc.nl; 2Netherlands Institute of Neuroscience, Royal Netherlands Academy of Arts and Sciences (KNAW), 1105 BA Amsterdam, The Netherlands

**Keywords:** retina, retinal pigment epithelium (RPE), adeno-associated virus (AAV), promoter, enhancer, polyadenylation, pro-viral plasmid, Cas9, transgene expression assay (TEA), biological activity assay (BAA)

## Abstract

Inherited retinal dystrophies and optic neuropathies cause chronic disabling loss of visual function. The development of recombinant adeno-associated viral vectors (rAAV) gene therapies in all disease fields have been promising, but the translation to the clinic has been slow. The safety and efficacy profiles of rAAV are linked to the dose of applied vectors. DNA changes in the rAAV gene cassette affect potency, the expression pattern (cell-specificity), and the production yield. Here, we present a library of rAAV vectors and elements that provide a workflow to design novel vectors. We first performed a meta-analysis on recombinant rAAV elements in clinical trials (2007–2020) for ocular gene therapies. We analyzed 33 unique rAAV gene cassettes used in 57 ocular clinical trials. The rAAV gene therapy vectors used six unique capsid variants, 16 different promoters, and six unique polyadenylation sequences. Further, we compiled a list of promoters, enhancers, and other sequences used in current rAAV gene cassettes in preclinical studies. Then, we give an update on pro-viral plasmid backbones used to produce the gene therapy vectors, inverted terminal repeats, production yield, and rAAV safety considerations. Finally, we assess rAAV transgene and bioactivity assays applied to cells or organoids in vitro, explants ex vivo, and clinical studies.

## 1. Introduction

Many reviews have been written on recombinant adeno-associated virus vector (rAAV) tropism in ocular tissue, rAAV host cell infection, and potential rAAV-treatable inherited retinal diseases [1,2,3,4,5,6,7,8,9]. Here, we will review the ocular gene therapies developed over the past 20 years focusing on the diversity of elements incorporated in rAAV vectors. Further, we will discuss how the vectors were generated, tested, and further modified to increase the potency and safety of the gene expression vector. After that, we will provide a library of validated elements that allows other researchers to streamline the modification of their vectors. Novel medical therapies, such as gene therapies, need to be carefully optimized to demonstrate their efficacy and safety before going into clinical human trials [10]. That is why, it is crucial to choose the most relevant biological model(s) (in vitro, in vivo, and ex vivo models) to test an optimized gene therapy vector in a transgene expression assay (TEA) and test the transgene activity in a biological activity assay (BAA). Thus, we will explore the developments in preclinical models, for example, cellular models, human induced pluripotent stem cell (hiPSC) derived retinal organoid disease models, retinal explant models, and tropism studies in non-human primates.

### 1.1. Why Viral Vector-Based Gene Augmentation Therapy for Ocular Diseases?

The environment of the eye offers a wide range of treatment possibilities due to the blood-retinal barrier of the eye decreasing viral vector diffusion to other organs and systemic immune activation. The retina consists of terminally differentiated cells thus reducing gene integration and chromosomal rearrangements. Many noninvasive techniques are available to monitor the treatment response.

Treatments aim to slow the progression of the inherited retinopathies by reducing retinal cell death, augmenting retinal function, replacing cells, or creating an artificial retina (retinal prosthesis). Preventing retinal cell death may be achieved by gene therapy, cell therapy, other drug treatments, as well as dietary adjustments and even by changes in lifestyle. Such as cigarette smokers giving up smoking or people taking too much high Vitamin E supplements [11]. Currently, there are only a few scientifically proven preventive or protective actions available to patients with inherited retinal diseases. Some preventative measures may potentially decrease patients’ quality of life, such as the continuous use of eye protection, photochromic lenses, or restriction to light exposure. Successful viral gene therapies can last many years. This could alleviate the disease burden thereby increasing the patients’ quality of life tremendously.

The ocular gene therapy strategy targets the basis of inherited retinopathies: the gene. With this strategy, the variant (disease-causing) gene is silenced, replaced, or repaired in the target cell (see Figure 1). The primary gene cassette carrier systems for ocular diseases has been the recombinant viral vector. However, many other potential technologies have been developed alongside, including exosomes/liposomes, antisense oligonucleotides (AONs), electroporation of naked DNA/RNA, or application of nanoparticles [12,13]. In most clinical retinal gene therapy studies, the vector of choice is the rAAV gene expression vector [14]. Two genes that do not fit in a conventional rAAV gene cassette (*MYO7A* linked to Usher syndrome Type 1B and *ABCA4* to Stargardt disease) have been delivered to the retina by recombinant lentiviral expression vectors instead [15,16,17,18]. Moreover, another large gene, *CEP290*, can be rescued on the mRNA level by antisense oligonucleotides (AONs; QR-110 for Leber congenital amaurosis (LCA), with *CEP290*; QR-421a for LCA with *USH2A*) [19]. An important consideration is how the therapeutic product is administered to the target cells. The most common method is the subretinal or the intravitreal injection. Subretinal injections can target a focal area (e.g., macula), favoring high vector delivery to the retinal pigment epithelium (RPE), photoreceptors, and Müller glial cells. Intravitreal injections efficiently target the ganglion cell layer and spread the rAAV to the whole retina in rodents but not in the primate retina due to the shielding properties of the inner limiting membrane. Figure 1 describes the considerations for choosing a gene therapy strategy.

### 1.2. rAAV Gene Therapy for Ocular Diseases—Advantages and Disadvantages

rAAV DNA carrier systems have been successfully used because they have a plethora of advantages: (1) They express the transgene within days or weeks and can reach full expression levels after 4–6 weeks in vivo [20,21]. (2) rAAV DNA carrier systems allow long-term treatment for at least ten years in dogs [22], (3) and primarily deliver their gene cassette in episomal concatemers into the nucleus [23]. (4) rAAV DNA carrier systems spread well within tissues to target large retinal patches [24]. (5) The capsid composition can be adjusted to fit one’s goals [25]. (6) rAAV injections have been very safe with a low number of serious adverse events (SAE) in clinical trials [26]. On the other hand, rAAVs similar to other viral strategies, have limitations such as: (1) A small gene cassette capacity (up to 4.6 kb + 2 × 145 bp ITRs). (2) The instability of the inverted terminal repeats (ITRs). (3) The need for high viral load for transgene expression. (4) A humoral immune reaction can be evoked. The immune reactions range from induction of neutralizing antibodies that reduce the number of capsids reaching the target cells to the innate immune pathways silencing the gene cassette within the host cell, as well as the cell-mediated T-cell immune response against foreign protein expression [27]. It is important to carefully consider the advantages of using rAAVs over other attractive strategies (Figure 1). In the following, we will describe how successful gene cassettes (vectors) have been designed for rAAV vectors (vectorology) that entered the clinical phase.

## 2. Ocular rAAV Vector-Based Therapies in Clinical Trials

Inherited retinal dystrophies (IRDs) and inherited optic neuropathies (IONs) are chronic and disabling disorders affecting 1/2000 to 1/4000 people worldwide. They display considerable genetic, symptomatic, and anatomical heterogeneity (Figure 2A; [28,29,30]). More than 250 genes can cause IRDs and IONs [31]. IRDs include pigmentary retinopathies, maculopathies, and stationary retinopathies. Patients with pigmentary retinopathies regularly suffer from night blindness, tunnel vision, and photophobia while maculopathies lead to the loss of color vision and accurate vision. Gene products worsening the ciliary development or the mitochondria function can cause syndromic IRDs. Common ciliary retinopathies are Usher syndrome, Bardet-Biedl syndrome, and Senior–Løken syndrome [32,33,34]. Some disease genes also affect the optic nerve causing vision loss by disturbing the electrical transmission from the ganglion cells to the cortex. The progressive degeneration of the optic nerve leads to ischemic optic neuropathies (IONs).

rAAVs are one of the most promising gene augmentation tools for the treatment of inherited ocular diseases. For instance, more than 33 rAAV gene therapies have been delivered to clinical trials (Table 1, Figure 2A). The current strategy of many ocular gene therapy trials is to shuttle and integrate a wild-type copy of the gene to the RPE or photoreceptor cells (Figure 2A). Figure 2A provides a list of genes intensively investigated for gene therapy. This basic rAAV gene delivery vector can consist of two inverted terminal repeats (ITRs) of the AAV serotype 2 (ITR2), a promoter, a wild-type copy of the cDNA of the gene-of-interest, as well as other enhancers or transcript stabilizing elements, an intron, and a polyadenylation sequence (Figure 2B). The promoter can be of viral origin, shortened-native, or synthetic. Many new promoters incorporate conserved transcription factor binding sites (TFBS), called enhancers, to boost transcription. Additional sequences can be added to a gene cassette, including fluorescent probes, linkers, base editors, nuclear localization signals, or short-hairpin RNAs (See Figure 2B and Section 3).

The first ocular rAAV clinical trial for *RPE65* was initiated in 2007. Over the years, five different AAV-*RPE65* products were tested in a total of 13 clinical trials by Applied Genetic Technologies Corporation (AGTC; Alachua, USA), Hadassah Medical Organization (Jerusalem, Israel), Spark Therapeutics (Philadelphia, USA), University of Pennsylvania (Philadelphia, USA), MeiraGTx (London, UK), Nantes University Hospital (Nantes, France), and University College London (London, UK). The clinical trial results lead to the first and only retinal gene therapy (so far) approved by the FDA in December 2017 and EMA in November 2018 (AAV-*hRPE65v2*; voretigene neparvovec-rzyl, LUXTURNA; Spark Therapeutics). The groundbreaking advancement was based on several approaches explored by different research teams concurrently. The five different strategies from the researchers delivered the *RPE65* cDNA to RPE cells by subretinal administration employing three different capsids (rAAV2, rAAV4, and rAAV5. Table 1) having different tropism and infection properties. Dose finding studies have shown that a high number of rAAV particles (>10^12^ viral genomes (vg)) can give rise to transient inflammation in mice [35]. This hurdle can be tackled by increasing the vector potency that reduces the dose and the danger of inflammation. Switching the capsid to rAAV5 increased the transduction of RPE (target) cells lowering the dose requirement. A lower dose is generally achieved by using a reliable, robust, and strong promoter that expresses (physiological) relevant levels of the transgene in the normal as well as diseased retina. Many gene therapies have employed the ubiquitous expressing viral CAG promoter that achieves high vector expression in the retina over many years. However, native promoters may permit a more cell-specific and natural expression profile. Two native promoters have been used for the rAAV-*RPE65* therapy, a 1.6 kb long native RPE65 promoter and later a promoter shortened to 750 bp (NA65p). The rAAV-NA65p-*RPE65* gene expression vector also had other modifications (SV40 intron; Kozak sequence; codon optimization) to increase potency and cell-specificity of *hRPE65* expression. The shortened NA65p promoter was much less silenced in the disease mouse retina than the longer promoter construct [36]. The search for the best product demonstrates the complexity of implementing native promoters (RPE65p, NA65p) over ubiquitous strong promoters (CAG, CB-SB) in transcription regulation over different animal models and disease states. We will discuss the different promoters and elements in Section 3. Currently, the two products, rAAV2/5-NA65p-OPTI*RPE65* and rAAV2/2-CAG-h*RPE65*v2 seemed to be at least as efficient in RPE65-deficient mouse retinal pigment epithelium [36].

The time it takes to move from a proof-of-concept study to a clinical trial is accelerating. Numerous breakthrough clinical phase I/II trials were already initiated over the recent years (see Figure 2A and Table 1) paving the way for therapies to come. Many more clinical trials, including for *CRB1*-related retinitis pigmentosa, will be initiated in the coming years. The acceleration is further exemplified by the shortened time it takes in moving from phase I/II to II/III clinical trials. ProQR moved from clinical trial phase I/II to II/III within one year (Product: AON QR-110 for *CEP290* mRNA). GenSight Biologics (Product: GS010, rAAV2/2-*ND4* for LHON) was able to move to clinical trial phase III within four years. NightstaRx Ltd. initiated the XOLARIS clinical trial phase I/II with a linked clinical trial phase III for Usher’s syndrome in which 200 enrolled patients in the study phase I/II could become included in the follow-up clinical trial phase III study (rAAV2/8-*RPGR*-ORF15). Furthermore, new products are constantly being developed, offering patients the hope of receiving a lasting therapy. Such as, age-related macular degeneration (AMD) that may be treated either by monthly administration of aflibercept (Product ProCon consists of *sFLT01*, an antibody-like product; Regeneron Pharmaceuticals) or by potentially long-lasting rAAV-therapy (Product: AAV2-*sFLT01*; Sanofi/Genzyme). Other exciting new ocular rAAV therapies are displayed in Figure 2A and Table 1. All clinical trial identifiers, clinical trial start dates, and products can be found in Appendix A.

The generation of the first recombinant AAV (rAAV2) was already described by the labs of Nick Muzyczka and Barrie Carter in 1982–1983 [37,38]. Many more AAV capsids have been found and engineered that are able to slow AAV capsid immune detection and degeneration and increase cell-specific transduction (see Section 5). So far, rAAV2 (and rAAV2 variants rAAV2-tYF and rAAV2-7m8), AAV5, and AAV8 have been injected into the human eye (Figure 3A). Evidently more clinical studies will follow with the sharp increase in rAAV clinical trial initiations since 2017. rAAV products moved from a few prototypes towards the use of a more diverse pool of cell-specific native and ubiquitous viral promoters (Figure 3B). Large promoters, such as the CBA/CAG (1661 bp), are less common in recent clinical trial products because of the limited packaging capacity of the rAAV. Now, new products contain relatively small viral or native promoters, such as the ubiquitous viral CAG/cytomegalovirus (CMV) promoter versions of less than 1 kb (truncated chimeric CMV/CBA promoter (smCBA); CB-SB; CMV; shortened CMV early enhancer element and a chicken β-actin promoter (CB7)). These short promoters increase the vector capacity tremendously (see Section 3).

Apart from the promoter and capsid, we searched the literature for enhancers, stabilizing elements (e.g., introns, splice donor/acceptors, Woodchuck hepatitis virus post-transcriptional regulatory element (WPRE)), polyadenylation sequences, pro-viral plasmid backbones, and production platforms. The documentation of this has been neglected in the literature. The enhancers mostly used in the clinical gene therapy vectors were: The cytomegalovirus (CMV) enhancer that is present in CMV and the CBA/CAG promoters and the interphotoreceptor retinoid-binding protein (IRBP) enhancer used in front of the hRS1 promoter (clinical trial NCT02317887). Enhancers can increase the transcription of the transgene (potency) reducing the viral load needed.

Moving on to the introns, common synthetic introns apart from native introns in the gene are rabbit β-globin intron with splice donor/splice acceptor (SD/SA; in CAG promoter), SV40 intron with SD/SA, human β-globin intron, and synthetic introns (e.g., 5′-splice donor of the first human β-globin intron and the 3′-splice acceptor of an intron of the immunoglobulin gene heavy chain variable region; Gene product: ADVM-022). The regulatory element Woodchuck Hepatitis Virus (WHP) Posttranscriptional Regulatory Element (WPRE) was included in 8 products (RST-001; scAAV2-*P1ND4v2*; GT005; rAAV2-*REP1*; RetinoStat; rAAV.*hCNGA3*; UshStat; rAAV2tYF-CB-*hRS1*; See also Section 3.6). The introns and the posttranscriptional regulatory elements can also increase transgene transcription.

Generally, the main choice for a polyadenylation (polyA) sequence in rAAV vectors are the effective bovine growth hormone (bGH) and the late SV40 polyA sequences (Figure 3C). New short (synthetic) polyadenylation sequences are needed to allow CRISPR/*Cas9* constructs to fit in a single rAAV such as in the clinical trial product of Allergan/Editas Medicine Inc to correct the *CEP290* gene in patients (product: AGN-151587/EDIT-101. See Section 3.7 polyadenylation and Section 4.3 CRISPR/Cas9). Today, many clinical trial initiators exist. Some companies efficiently acquired new potential therapies such as MeiraGTx and HORAMA (Figure 3D).

Since relatively little information is provided in the literature, we also compared the different plasmids and production cell lines required to produce rAAVs for clinical trials. Most rAAVs were produced in HEK293(T) cells without the use of helper viruses except for the products tgAAG76 (B50 cell line and helper adenovirus; [39]), rAAV2/2-*ND4* (HEK293 infected by HSV1-rc/ΔUL2; [40]), rAAV2tYF-CB-hRS1/rAAV2tYF-PR1.7-*hCNGB3*/rAAV2tYF-GRK1-*RPGR* (sBHK cells infected with rHSV; [41,42,43]), and ADVM-022 (Baculovirus Sf9; [41]). A list of the pro-viral plasmids of clinical trials can be found in Appendix A. A preliminary study investigated if the choice of the production cell line might influence the tropism and potency of the rAAV vector [44]. It showed that the rAAV capsids can have post-translational modifications, such as glycosylation, that depend on the species origin of the production cell. Furthermore, rAAVs produced in a human cell line (HEK293T) compared to baculovirus-*Sf9* produced rAAVs were more potent in transfecting the liver in mice in vivo and in vitro (HEK293T, Huh7, human induced Pluripotent Stem Cells (hiPSCs), primary human fibroblasts, mouse C2C12 cells). Further studies will need to address what specifically caused the increase in the transduction efficiency: the post-translational modification, species-dependent impurities in the rAAV preparation or the difference in the general production process between HEK293T cells compared to the baculovirus? A more detailed description of pro-viral plasmids for the production of rAAVs for clinical trials is needed to move towards safer plasmids (more information in Section 4.4.1: Production and Appendix A).

## 3. Discovery of Cell-Specific Promoters for Ocular Gene Therapy

### 3.1. Core Promoters in rAAV-Vectors

Eukaryotic RNA-polymerase II-dependent promoters consist of a core promoter and cis-acting regulatory elements that can include enhancers and silencer motifs [45]. In humans, the cis-acting regulatory domains and core promoters frequently contain cytosine-phosphate-guanosine islands (C:G ratio >60% for >200 bp). Recent studies indicated that the reduction of CpG islands in rAAV vectors increased transgene expression and reduced TLR9-mediated innate immune detection [46,47].

Minimal/core promoters require a transcription start site (TSS), a sequence motif for general transcription factors (e.g., TATA-binding protein or TFIIB) directing the binding of the RNA-polymerase II (e.g., ~35-bp upstream positioned TATA/CAAT/GC-box sequence) [45]. Many genes have more than one TSS that are differentially active in tissue and at various developmental stages. Therefore, many rAAV vectors with different promoters are produced and screened on transcription activity in retinal cells. Core promoters are often fused with (predicted) cell-specific enhancer and suppressor fragments. The core promoters of strong ubiquitous promoters (e.g., core CMV [30 bp], SV40mini [106 bp], SCP3 [81 bp] Table 2) have been extensively used in promoter element library screens [48,49,50].

### 3.2. Ubiquitous Promoters in rAAV-Vectors

Most of the promoters used in rAAV vectors are unidirectional ubiquitous promoters such as the cytomegalovirus early enhancer/cytomegalovirus promoter (CMV), the minimal CMV promoter (~300 bp), the cytomegalovirus early enhancer/chicken β-actin promoter (CBA aka CB7; 800 bp), the cytomegalovirus immediate-early enhancer/chicken β-actin promoter/rabbit β-globin intron (CAG or CAGGS or CBA), the human phosphoglycerate kinase (PGK) promoter, and the elongation factor-1 alpha (EF-1α) promoter [51,52,53,54,55]. The CAG, CMV, and CBA promoters outperform the EF-1α and PGK in total expression in the retina [54]. Smaller derivatives of the promoters have been developed with comparable expression patterns in some but not all tissues, such as the CMV/CBA-derivative CMV early enhancer with the chicken β-actin promoter with a chimeric chicken β-actin minute virus of mice (MVM) viral capsid protein (VP1) intron (CBh; ~800 bp), the CBA-derived CMV early enhancer with the chicken β-actin promoter and a truncated SV40 late 16S intron (CBA aka CB7, ~800 bp), and the minimal CMV promoter (~260 bp [24]; see Table 2). However, several ubiquitous promoters are silenced in specific cell types and tissues. For example, the CMV promoter had a sharp onset of expression but was silenced compared to the CBh promoter over ten weeks when expressed in the hippocampus, the spinal cord, or the substantia nigra. In contrast, the CMV promoter was not silenced in the striatum [54,56,57,58]. The role of CMV cis-regulatory silencing in the retina is less established. Administration of CMV.*eGFP* DNA incorporated into nanoparticles showed robust expression in the retina after two days, but the expression was not detectable after two weeks [59]. We have found protein expression (GFP, CRB1, or CRB2) in photoreceptors after one to three months of rAAV vector administration in wild-type and *CRB1*-related retinitis pigmentosa mouse models [24,60]. *SpCas9* was also detected in retinal flat mounts in mice two weeks post-subretinal injection of the rAAV-CMV.*spCas9* [61]. Similarly, the expression of GFP was detected after two to four weeks in human iPSC-derived retinal organoids transduced with rAAV-CMV-*eGFP* vectors [62]. The studies described above indicate that the ubiquitous CBA and CMV promoters are most likely less affected by the retinal disease state or cellular differentiation status.

### 3.3. Bicistronic and Tricistronic Promoters in rAAV-Vectors

Expression of two or more genes in a gene therapy vector can be achieved with an Internal Ribosomal Entry Sequence (IRES; non-read through linker), or otherwise, at least two different promoters could be used. The promoters could drive the expression of multiple genes with a fusion protein linker (e.g., (Gly_4_ Ser)_2_ spacers (~30 bp; read through linker), or a sequence encoding self-cleaving peptides (T2A, P2A, E2A, F2A; ~30–75 bp; read-through linkers) between the gene sequences. Fusion proteins can, however, alter the function of some of the proteins. The cleavage efficiency of genes connected with self-cleaving peptide sequences varies. Moreover, self-cleaving peptide sequences add additional amino acids that stay attached to the protein product. Moreover, several genes from one promoter connected with a linker generally reduce the expression of each subsequent gene. Nevertheless, researchers demonstrated the feasibility to mediate expression by a single promoter of three different genes (for example, *Oct4*, *Sox2*, and *Klf4*) connected with self-cleaving peptide sequences in a rAAV9 expression vector upon transduction of ganglion cells. The rAAV9-*Oct4*/*Sox2*/*Klf4* expression vector rescued ganglion cell survival in an optic nerve crush mouse model [71].

The interspersing IRES (572 bp) or minimum IRES (436 bp) allows efficient expression of independent genes into cap-independent RNA transcripts [72]. Yet, studies indicate a decrease in protein production of the protein-coding DNA located behind the IRES compared to the use of a conventional promoter. Adding a spacer (~30–90 bp) in the inter-cistronic sequence can enhance the IRES-dependent translation of the second gene [73,74]. Many different IRES exist that have been extracted from different viruses, such as in the family of the picornaviridae. Placing two promoters in opposing directions next to each other also allows efficient bicistronic gene expression from one gene cassette [75]. Bi- or tricistronic rAAV gene cassettes are especially useful where the protein of interest (e.g., *Cre*-recombinase) is expressed in a specific cell type together with an internal marker (e.g., for reporter gene assays), or when studying retinal circuits (e.g., by calcium imaging), or when performing rAAV-retrograde labeling [52,76,77,78,79].

Bicistronic rAAV gene cassettes hold the key to supplement a wild-type transgene and removing disease-causing variant proteins in an all-in-one rAAV vector therapy. For example, in autosomal recessive retinal disease, ocular gene augmentation therapies express a functional gene in retinal cells that lack a functional copy of that same gene. However, in autosomal dominant (e.g., rhodopsin) or X-linked dominant (e.g., some variations in *RPGR*) retinal diseases, the allele bearing the dominant-negative variation needs to be ideally inactivated. Some dominant diseases do respond positively to gene augmentation, but inactivation might further slowdown vision loss. Such inactivation can be achieved, for example, by gene editing or small interfering RNA to allow gene augmentation therapy to work. In the latter case, to prevent inhibition of the newly introduced gene, codon-optimization of the transgene might prevent inactivation by the gene-editing or siRNA tools used [80]. Here, an rAAV vector expressing a wild-type *RPGR* transgene and downregulation of the mutant *RPGR* transgene could benefit patients.

Similarly, many inherited retinal diseases benefit from the administration of cell survival factors [81,82,83,84,85]. The expression of cell survival factors, such as the basic fibroblast growth factor (*bFGF*; 470 bp), ciliary neurotrophic factor (*CNTF*; 600 bp), glial cell line-derived neurotrophic factor (*GDNF*; 511 bp), pigment endothelial-derived factor (*PEDF*; 883 bp) and brain-derived neurotrophic factor (*BDNF*; 750 bp) can be expressed concomitantly with the gene-of-interest boosting the treatment effect [86,87]. Combining a gene supplementation therapy with a supporting factor expressed from one rAAV vector is very promising for future treatments.

### 3.4. Retina-Specific Promoters

The use of these native occurring regulatory sequences may actively modulate transcription and thereby preventing overexpression. A native promoter could, therefore, potentially reduce toxicity due to overexpression of the transgene. Cellular toxicity can, for example, be observed in rAAV shRNA overexpression studies in which ubiquitous promoters were used that caused saturation of cellular miRNA pathways [88].

Many retinal cell-type-specific promoters have been developed (Table 3). The selection and validation of tissue-specific promoters can be complicated and time-consuming. Many tissue-specific promoters in mice turned out to be less specific in human or non-human primates [50]. Further, many tissue-specific promoters drive much lower gene expression, compared to the CBA/CMV/CAG ubiquitous promoters. Nevertheless, many tissue-specific promoters can efficiently express many transgenes from rAAV-gene cassettes: NA65p (RPE cells), Nefh (ganglion cells), hGRK1 (rod and cone photoreceptor cells), hRLBP1 (Müller glial cells and RPE cells) and others [24,36,89,90,91]. For an extensive list, see Table 3. cDNA codon-optimization and the inclusion of introns (e.g., MVM, SV40) and enhancers (e.g., CMVe, IRBPe, Grm6e) to tissue-specific promoters can substantially increase the promoter activity (see Section 3.6). For example, the NA65p promoter is derived from the hRPE65p but now has a 150× higher potency than the CBA in mice [36]. The optimized promoter may become useful to efficiently express transgenes (e.g., CRISPR/Cas) in the normal retina, but it is not yet proven that the promoter works as good in the diseased human retina. The promoter might be less active or overactive in diseased retinal target cells that express the required transcription factors at other levels than in the normal retina.

Cell-specific promoter activities are inherently difficult to predict how they will fare in disease states when the pool of transcription factors in a cell change. Many viral promoters evolved to maximize the survival of the virus in different cellular contexts. Several viral promoters exhibit expression in many cell types in various cell “states” (stressed, developmental state, cell cycling). For example, the transactional activation of the widely used CMV or early SV40 promoters is related to the binding of the ubiquitous transcription factors Sp1/Sp3 [92,93]. Moreover, most mammalian cells express the Sp1 transcription factor [94]. Adding the CMV immediate-early enhancer (CMVe or CE) to cell-specific promoters generally increases expression but may also decreases specificity (see Table 4).

Unfortunately, many tissue-specific promoters are too large to fit into rAAV vectors. Fitting depends on the size of the gene of interest, which is why many promoters are further shortened and optimized for cell-type-specific expression. We reduced the length of a Müller glial cell-specific CD44 promoter from 1775 bp to 363 bp but then abandoned the shortened CD44 promoter because of a substantial loss of expression in vivo [24,95]. The full-length glial fibrillary acidic protein (GFAP) promoter (2789 bp) showed excellent Müller glial specific gene expression in human retinal organoids and human retinal explants [24,62]. Furthermore, a shortened version of the GFAP promoter called gfaABC1D (686 bp) showed similar expression strength in neurons (brain), whereas the gfaABC1D promoter maintained Müller glial cell-specific expression in the retina [96,97]. We compiled a list of relatively short promoters that have been at least tested in the mouse retina (Table 3). The list might allow to further find enhancer elements to generate novel short, specific, and strong promoters for ocular gene therapies.

### 3.5. Small Nuclear RNA (snRNA) Promoters

RNA polymerase (RNAP)-dependent regulatory promoters (U1, U2, U6, U7, H1; ~250 bp) can be used to drive short hairpin RNAs (shRNAs). The human U6 promoter is a potent promoter that has been widely used for the expression of shRNAs. However, the relatively large size, the requirement that the transcript starts with a G or A, the sometimes too active transcription, and the sensitivity to specific cellular profiles make the human U6 promoter a less versatile promoter [88,131,132]. RNA polymerase III promoters also have been re-engineered with CMV enhancers [133] or tissue-specific enhancers (heart, muscle) for siRNA expression [134]. The tissue-specific enhancers increased expression but were less tissue-specific.s Single guide RNAs (gRNA) are typically expressed by a U3 or U6 RNA promoter in rAAV gene cassettes. Relative tissue-specific expression of two gRNAs for CRISPR/Cas gene editing in myotubes was achieved by linking a muscle-specific MHCK7 promoter (pol II) with gRNA-linked self-cleaving ribozyme sequences derived of Hepatitis delta virus (HDV) and a Hammerhead (HH) sequence [134,135,136,137]. The research for promoters for tissue-specific expression from a small nuclear RNA promoter is an underrepresented field. More promoters to choose from would increase the number of genes that target for ocular gene therapy. Currently, many gRNA rAAV gene cassettes circumvent the lack of snRNA cell-specific promoters by driving the expression of the gRNA from ubiquitous snRNA but the Cas9 from a cell-specific promoter (see Section 4.3 for examples).

### 3.6. WPRE, Introns, miRNAs, and Other Elements in a rAAV-Gene Cassette

Post-transcriptional regulatory elements (PRE) can substantially increase gene expression. Woodchuck posttranscriptional regulatory element (WPRE; 600 bp) or Hepatitis B Virus Posttranscriptional Regulatory Element (HPRE; 533 bp) increase the transgene expression up to 6–9 times [57]. The addition of a WPRE also protects from silencing in human ES cells and the brain. To validate the use of WPRE for retinal gene therapy, rAAV2/2.CMV.*eGFP*.pA vectors with or without WPRE were applied to human retinal explants or injected in mouse eyes [138]. A shorter version of the WPRE (WPRE3; 247 bp) showed only a 15% drop in expression in hippocampal neuron cultures or GFP expression in rAAV infected hippocampal CA1 region in the mouse brain [139]. A modified WPRE version that removed any viral protein expression has been patented for retinal use [140]. WPRE might be redundant if used in combination with a promoter containing introns. No increase in transgene expression was found when added to the CAG or EF-1α indicating that a maximum of transcriptional activity and quality can be reached by having good regulatory elements found in certain promoters containing introns or in the WPRE [75,141]. Inclusion of natively occurring or synthetic introns can strongly boost protein expression, especially for vectors with low efficiency of gene splitting sites [142]. Many introns have been developed for rAAV gene cassettes that can enhance the gene expression (Table 4). Especially, the strong MVM intron-1 of the viral capsid protein (*VP-1*) of only 67-to-97 bp can increase the transcript expression by 10× [143]. Moreover, the development of minicircle rAAVs has contributed to novel introns that are placed in the backbone of pro-viral plasmids to boost production yield. This strategy will be discussed further under section production [144].

Adding microRNAs (miRNA, ~18–25 bp) can alternatively be used to prevent the ectopic expression of the transgene in ocular gene therapy. Adding 4× the complementary sequence of miRNA204 to a rAAV2/5.CMV.*eGFP*.WPRE.*4xmiRNA204T* significantly reduced eGFP expression in RPE cells after subretinal injection in mice and pigs. Similarly, adding 4× the complementary sequence of miR-124 removed the expression in photoreceptors [145]. Furthermore, a dual-acting rAAV2/5 vector expressed the miRNA (5, B, 7), against Vascular endothelial growth factor A (VEGFA) and antiangiogenic protein pigment endothelial-derived factor (PEDF) driven by an RPE-specific Bestrophin 1 (VMD2) promoter, suppressed choroidal neovascularization in a wet-AMD mouse model [87]. However, the oversaturation of the cognate miRNA needs to be considered when using miRNAs, because they can decrease the function of native miRNAs in the cell. 4× miRNA placed in an rAAV-CMV expression cassette generally is sufficient for miRNA expression without inducing side-effects [145]. Others have used miRNAs to inhibit transgene expression in antigen-presenting cells (APCs) with miR-142-3p [146]. Still, short hairpin DNA sequences need to be placed at least proximal to the second ITR and be tested for possible rAAV genome truncation for proper expression of the short hairpin RNA (shRNA). Short-hairpin DNA can effectively truncate rAAV genomes during production and produce non-intact shRNA expression cassettes [147]. A more detailed review of miRNAs can be found here [148].

### 3.7. Polyadenylation Sequences in rAAV-Gene Cassette

To allow for efficient pre-mRNA processing, an efficient polyadenylation sequence needs to be included behind the transgene to form a proper poly(A) tail at the RNA’s 3′ end. Polyadenylation sequences in rAAVs gene cassettes are listed in Table 5 and include for example SV40 late (135 bp; +++), bGHpolyA (250 bp; ++), synthetic polyadenylation (spA) + 2× SV40 late upstream elements (100 bp, ++), 2× sNRP1 (34 bp, +/++), synthetic polyA (spA; 49–60 bp, +), hGHpolyA (624 bp, +), 1× sNRP1 (17 bp, +), and adenovirus L3 (21 bp, +) polyadenylation sites [57,165].

Recent developments allow for shorter and more potent expression cassettes. The SV40 late polyadenylation signal upstream element and the SV40 late polyadenylation signal combined with the WPRE3 (420 bp), decrease the length to less than half compared to the commonly used WPRE-bGHpolyA gene cassette (919 bp), and maintain a similar expression profile [139]. The removal of a WPRE sequence reduced the expression by 80%, whereas the use of a synthetic polyadenylation sequence (49 bp) in combination with two SV40 late upstream elements (50 bp) increased the GFP expression compared to the use of a robust bGHpolyA sequence. Interestingly, the interplay of the polyadenylation sequence with transcriptional regulation enhancers can increase transcript levels. However, the effect can be quite specific to elements. For example, paring a CMVβ enhancer with an SV40 polyA increased transcript levels but the CMVβ enhancer paired with a bGHpolyA did not [149].

Also, the rAAV gene cassette for hemophilia B was tested with different polyadenylation sequences. The bGHpolyA turned out to be the strongest for the *FIX* gene expression, outperforming other polyA sequences such as the synthetic polyA, mouse β-globin pA, rabbit β-globin pA, and H4-based pA [143]. Studying polyadenylation sequences can be very valuable for rAAV gene cassette size reduction. Notably, a 17-bp soluble neuropilin-1 (sNRP-1) polyA sequence efficiently expressed transgenes on infection of an rAAV vector. When the sequence was used twice (2× sNRP-1 polyA), then the potency was as efficient as a single SV40polyA sequence [165,166]. Yet, the two sNRP-1 polyA were less suitable for specific transcripts compared to bGHpolyA or spA [167]. The effects of polyadenylation sequences for specific transcripts are still less well understood. For example, whereas polyA’s increase transcript stability/expression, certain polyadenylation sequences can also reduce viral titers during rAAV particle production [168]. Thus, different polyadenylation sequences should be tested for optimal gene expression and virus production.

### 3.8. rAAV Vector Cassettes and Inducible Promoters

Many gene supplementation therapies rely on constant overexpression of the therapeutic gene. The constant active expression increases the risk that the rescue vector itself becomes toxic to the cell. Stress (GFAP promoter) or hypoxia-driven GFAP promoters (HRSE-6xHRE-GfaABC1D) have been generated that might be safer for cells that are sensitive to continuous overexpressed artificial gene vectors [96,97,99,100,102]. Other inducible On/Off gene expression systems have been described: Tetracycline (Ptet), dihydrofolate reductase (DHFR) protein destabilizing domains, riboswitches, metal activated promoters (metallothionine-Ia; MT-1), and hormone-activated promoters (dexamethasone, MMTV LTR. Table 6) [173,174,175,176]. All but the riboswitches require the expression of an exogenous (bacterial) protein. The TET-system is activated by an antibiotic (tetracycline or doxycycline), making it suboptimal for human use. An example of an efficient TET-off rAAV system is an rAAV expression cassette that includes 6x the mutated tetracycline response elements (TRE; ~200 bp) placed in front of a minimal promoter (CMV; ~40 bp; total cistronic size: ~270 bp). Upon rAAV infection of the cell, then the ubiquitous promoter UbC will drive the transactivator reverse tetracycline transactivator 3 (rtTA3), making it a Tet-on system. Upon Cre recombinase expression, the rtTA3 is floxed-out, rendering the expression vector to a Tet-off system. The rtTA3 binds to the TRE in the presence of doxycycline, starting the expression of the TurboRFP open reading frame (ORF) that allows tracking of the target mRNA knockdown because the miR-30 sequences induce the Drosha and Dicer processing of the expressed target sequence. The promoter drives the micro RNA adapted short hairpin RNA. If the Drosha/Dicer degradation complex recognizes the target sequence specified by the shRNA, then the transcript of the target sequence and the TurboRFP transcript is degraded. The construct allows fast testing of the efficiency of shRNAs [177].

Riboswitches have gained considerable attention for rAAV ocular therapies because of their small size (100 bp cis-acting RNA sequence), adaptability to ligands, and the development of synthetic riboswitches [173,178,179]. The riboswitch encodes for a ligand-sensing aptamer, a communication module (linker), and an effector domain (ribozyme) that depending on the presence of the ligand, cleaves the mRNA of the gene expression cassette. A proof-of-concept-study for anti- vascular endothelial growth factor (VEGF) expression by the activating ligand tetracycline in a wet AMD mouse model demonstrated the feasibility of the riboswitch in ocular gene therapy [179].

## 4. Optimizing Genes for rAAV Vector Therapies (Minigenes, Dual/Triple rAAV-Vector, ITRs)

### 4.1. Intron Removal, Exon Removal, Surrogates, and Pathway-Modifying Therapies

rAAV transgene cassettes only allow expression of small genes because of the limited packaging size of around 4.5–4.6 kb (excluding the two ITRs). It has been challenging to fit large genes into rAAV vectors. Introns and sometimes exons are truncated to allow proper packaging. Thus, many large cDNAs in rAAV gene cassettes do not contain any or contain only a few native introns (see Table 1, and Appendix A). Introns can serve many functions, for example, to increase mRNA stability, modulate RNA synthesis rate, introduce alternative splicing, and decrease DNA damage of highly expressed genes. Moreover, an intron can dictate the mRNA export mechanism from the nucleus to the endoplasmic reticulum [181,182]. Removal of introns in rAAV gene cassettes might further alter the intrinsic gene regulation apart from artificial promoters and rAAV expression system. For example, the CRB1 protein (UniProtKB P82279-1) consists of 1406 amino acids and is encoded by 210,251 bp (GRCh37 197,237,334-197,447,585. ENSG00000134376). The exon-coding sequence alone is 4221 bp, including the 3′ stop codon, whose size is close to the maximum packaging capacity of the rAAV. Nevertheless, we achieved an efficient expression of CRB1 in retinal cell types by making use of short promoters (<300 bp) and a 50 bp synthetic polyadenylation sequence [60].

Shortened versions of proteins (exon truncations aka *minigenes*) are generally not advised because many shortened proteins lose their functionality. For example, many shortened versions of the Duchenne muscular dystrophy gene (*DMD*; 2.3 Mb; 79-exons; 3685 amino acids; 11,055 bp) do not rescue the Duchenne muscular dystrophy phenotype in muscle cells except for the micro-dystrophin for which was reported a milder clinical phenotype [183,184]. Nevertheless, the micro-dystrophin rescue in patients in clinical trials has been at least suboptimal and needs further optimization [185]. We also tried a native occurring short version of the CRB1 protein encoded by a *CRB1* cDNA lacking exons 3 and 4 while maintaining the open reading frame, but whereas the short *CRB1* (*sCRB1*) was expressed, it also caused retinal degeneration upon subretinal injection of the AAV9-CMV-*sCRB1* or AAV9-hGRK1-*sCRB1* [24]. Moreover, the *CEP290* gene (7.5 kb cDNA coding for 2479 amino acids) is too large to fit in a rAAV vector. A smaller version of the *CEP290* cDNA (minigene mini*CEP290*^580–1180^ coding for the RAB8A-binding domain) temporarily rescued in part the ciliary length and retinal function in the *Cep290*^rd16^ mouse [186]. The company Ophthotech (now Iveric) explores the mini*CEP290*^580–1180^, a mini*ABCA4* (no published information available), and mini*USH2A* (no published information available) for rAAV vector therapy. The *USH2A* gene (15.6 kb cDNA coding for 5202 amino acids) cDNA has been shortened to the mini*USH2A*-1 (~6.8 kb) and mini*USH2A*-2 (~4.1 kb) and delivered by *Tol2* transposase mRNA into homozygous one-cell staged *ush2*^armc1^ zebrafish embryos restoring visual motor responses and retinal function [187].

Instead of employing a shortened protein, one can also apply a surrogate protein such as utrophin for dystrophin [185,188]. We also developed a surrogate gene therapy for patients with *CRB1*-related retinal dystrophy by employing the Crumbs homolog 2 gene (*CRB2;* 3.8 kb). However, such a strategy can only be performed if the proteins of interest execute similar functions in cells, which is the case for CRB1 and CRB2 [62,189,190,191,192,193,194,195,196,197,198,199,200,201,202]. The rAAV-CMV-*CRB2* rescued the loss of CRB1 function in mouse Müller glial cells [60]. Interestingly, mice lacking CRB1 or mouse retinas lacking CRB2 develop the same phenotype as observed in human iPSC-derived retinal organoids lacking CRB1 [62]. Surrogate proteins might be less immunogenic than the native protein—such as utrophin over dystrophin—because the surrogate protein is already expressed in the body. Surrogate proteins could have great potential for many rAAV gene therapies.

The identification of key players in signaling pathways (e.g., VEGF, TGF-β, Wnt) is an exciting but complex research area. Here, exemplified on glaucoma therapies, the expression of a key player is altered (e.g., by antibody or viral vector administration) to induce a lasting change on intraocular pressure (IOP). The search for key players is generally on: (a) mutated genes that are more frequently found in patients with glaucoma (single nucleotide polymorphisms [SNP] databases; e.g., *CAV-1*), (b) genes affected by glaucoma (e.g., data from [single-cell] RNAseq healthy-disease condition databases; e.g., *ANGPTL7*, *MMP1*, and *PLAT*), or (c) gene products that are actively involved in regulating the intraocular pressure (e.g., data from knock-out/in gene library screens and literature searches; e.g., RhoA-Rho kinases and prostaglandin EP4 agonists) [203]. Similar approaches are explored for the identification of pathway genes for RP and LCA.

The expression of pathway-modifying factors can be done by the injection of neuronal progenitor cells that release growth factors (jCyte, Inc, product jCell; ReNeuron Limited, product hRPCRP) or viral vectors expressing the specific key player (see Figure 1, Table 1 rAAV products for AMD/glaucoma). Cell survival factors, such as pigment-endothelial-growth factor (PEDF), are discussed in the Section 3.3. For example, the suppression of the vascular endothelial growth factor receptor (VEGFR) activation and the complement cascade activation in the RPE and choroid by repeated intravitreal antibody injections is successfully used in clinical studies [204]. This led to the development of rAAV gene cassettes that express upon infection a soluble protein fragment that partially binds to the VEGF receptor or complement system proteins inhibiting the disease pathway cascade. The vectors have efficiently prevented a full-blown AMD phenotype in AMD models and are currently tested in clinical trials: (1) the small soluble fms-like tyrosine kinase-1 (*sFlt-1*. Non-membrane associated splice variant of VEGFR1 encoded by the *FLT1* gene; Adverum Biotechnologies; Sanofi Genzyme) [41,205,206], (2) *endostatin* (cleavage product of collagen XVIII)/*angiostatin* (cleavage product of fibrinogen; Oxford Biomedica) [207], (3) the complement factor I (CFI, a C3b/C4b inactivator in the complement cascade; Gyroscope Therapeutics) [208], (4) anti-VEGFfab Heavy chain and anti-VEGFfab Light chain (Regenxbio) [209], and (5) the soluble CD59 antigen (binding to C5b preventing C9 incorporation in the complement cascade; Hemera Biosciences) [210].

Finally, the expression of light-activated opsin-like proteins are explored to restore (partial) vision in RP patients (see Table 1 rAAV products RST-001, GS030, and BSO1). Three different fluorescent proteins (ChR2 aka Channelrhodopsin-2 [930 bp], ChR88m19-tdTomato aka ChrimsonR [2496 bp], and Chr90 aka Chronos [1710 bp]) are explored in clinical trials [211,212]. Chronos-GFP (green-shifted) and ChrimsonR-tdTomato (red-shifted) are second generation fluorescent proteins that have faster kinetics and are more light sensitive compared to ChR2. Further tests are needed to find out which approach restores optimal vision in patients with ocular diseases: (1) a wild-type or surrogate gene (e.g., *CRB1* or *CRB2* to a *CRB1*-RP patient), (2) a shortened gene supplementation (e.g., mini*CEP290*^580–1180^), (3) neuroprotective factors (e.g., *CNTF*), (4) the delivery of antiangiogenic factors (e.g., sFLT1), or (5) disease pathway-modulation (e.g., *ANGPTL7* in glaucoma), or (6) the introduction of a light-activated protein. We will mainly focus on the supplementation of a wild-type copy of a gene for the rest of the review.

### 4.2. Lentiviral and Dual/Triple rAAV Vectors

Retroviral 3rd generation lentivirus-based systems have a larger packaging size of ~8.5 kb compared to 4.5–4.6 kb in rAAV [213]. They can infect both dividing and non-dividing cells and integrate into the genome. Equine infectious anemia virus (EIAV) 3rd generation lentivirus-based gene therapies for the *MYO7A* gene (6645 bp; 2215 amino acid) and the *ABCA4* gene (6819 bp; 2273 amino acids) delivered to photoreceptors have been in clinical trials for Usher syndrome type I and Stargardt disease (NCT01367444; NCT01505062) since 2011/2012, respectively. Lentiviral vectors efficiently infect RPE but it is less clear how efficient these vectors are on photoreceptors. The results of the clinical gene therapy studies might tell us whether or not the EIAV lentiviral approaches onto photoreceptors have been efficacious.

Dual and triple rAAV vectors are another strategy to circumvent the small capacity of rAAVs. Nicked ITRs (∆ITRs) have been used that allow for annealing of two or three different rAAV gene cassettes. An update on dual and triple rAAV vectors can be found in a recent paper by Trapani [214] who successfully applied a dual *ABCA4* AAV vector system in *Abca4* knockout mice [215,216]. The first generation of dual rAAV vector cassettes resulted in a high ratio of truncated gene expression. Adjustments, such as 200–300 bp of specific compatible overhangs, have resulted in normal concatemerization of independent gene cassettes such as the hybrid dual rAAV approach [214].

### 4.3. rAAV-Vectors Expressing CRISPR/Cas

*Staphylococcus aureus* CRISPR associated protein 9 (SaCas9) is an RNA-guided endonuclease enzyme associated with the CRISPR (type II prokaryotic Clustered Regularly Interspaced Short Palindromic Repeats) complex. Cas9 unwinds, checks, binds, and finally cuts in the DNA (causing a double-stranded DNA break [DBS]) complementary to the annealed 20-nucleotide genome-specific part of the single guide RNA (gRNA). The genome-specific part of the gRNA anneals proximal to the 3-bp protospacer adjacent motif (PAM). The guide RNA can be adjusted to target the whole genome as long as a PAM sequence is found close by (for *S. aureus*: NGG). Many Cas protein homologs and orthologs have been described with the most significant ones for rAAV gene-editing cassettes being Cas9, Cas12a (Cpf1), Cpfl1, SpCas9, SaCas9 [217]. The large Cas9 (*SpCas9*, 4100 bp) or type-V Cas system (*AsCpf1*, 3921 bp; *LbCpf1*, 3684 bp) together with the gRNA cassette generally do not fit smoothly in a single rAAV gene cassette. The new generation of *SaCas9*, *CjCas9*, and *NmCas9* (2.9–3.3 kb) allows the packaging of both Cas9 and gRNA in a single AAV vector. CRISPR/Cas gene editing can inactivate the dominant-negative effect or can regulate positively or negatively the transcription of genes. However, if left active in cells, functional rAAV-CRISPR/Cas9 systems do increase the number of off-target integration events into the genome [218].

The large *SpCas9* (4.1 kb) would require a dual rAAV system to incorporate all elements, including the gRNA cassette. A dual rAAVs system (rAAV.RKp.*SpCas9*; rAAV.U7.*gRNA-Nrl*) rescued vision in three mouse lines of rod retinal degeneration (*Crx-Nrl^-/-^*; rd10 or *Pde6b^-/-^*; *Rho^-/-^*) by knocking out *Nrl* in one or both alleles [219].The *Nrl*-knockout pushed rod photoreceptors to a more cone-like state helping in the survival of the remaining photoreceptors. Similar results have been reported in a second independent study [220]. The mutant rhodopsin gene encoding a dominant-negative form of rhodopsin (*Rho^P23H/P23H^*) was also silenced by gene editing in a mouse model of retinal degeneration (*Rho^P23H/P23H^*) by a dual rAAV-vector administration (rAAV2/8(Y733F)-sCMV-*SpCas9.*spA and rAAV2/8(Y733F)-U6.*gRNA1gRNA2(mRho).*mRho.h*RHO.*SV40-polyA) rescuing retinal degeneration [61].

A shorter Cas protein, *CjCas9* (2950 bp), allows expression from a single rAAV vector. Intravitreal injection of a single rAAV2/9-vector at P0 in mice efficiently downregulated angiogenesis genes in mice (rAAV-gRNA against *Vegfa:Hif1a.CjCas9*-T2A-*GFP*). The downregulation was linked to a decrease in spot size on morphology of the laser-induced choroidal neovascularization (wet AMD model) but not on retinal function measured by electroretinogram (ERG) [221]. Their follow-up paper showed that 14-months post-injection, the *CjCas9* is still active but does not affect the retinal function indicating that the therapy might be safe [222].

Cas proteins can also be altered and fused to other proteins. For example, the 3.2 kb cDNA encoding a nonfunctional nuclease-activity-dead *S. aureus* ortholog Cas9 (dCas9) can be fused with a cDNA encoding a transactivation domain such as VP64 fused to the two transcription factors p65 and Rta (*dCas9-VPR*). The cDNAs encoding VP64, p65, and Rta are 150 bp, 357 bp, and 570 bp in length, respectively. Because of the relatively large size of 4277 bp of *dCas9-VPR* cDNA, the authors used a dual rAAV system to express the *dCas9-VPR* and gRNA expression cassettes [223,224]. Recently, a single rAAV expression vector has been developed driving the gRNA by a U6 promoter (360 bp) and a shortened but 3× less active *VPR* (500 bp) and *dSaCas9* (3200 bp) from an SCP1 promoter (80 bp) attached to 2×-sNRP-1 polyA signal (34 bp). A modified rAAV-vector version with a full-length CMV promoter and the bGHpolyA efficiently upregulated a gene (*Actc1*, *Neurog2,* or *Hbb*) upon infection of N2A neuron derived cells by 50–150× in vitro [167]. A single rAAV vector expressing dCas9 fusion protein, as well as a sgRNA, shows excellent potential for positive or negative regulation of transcription in many genetic pathways involved in retinal diseases. Several other exciting *Cas9* gene cassettes in rAAVs will most definitely be developed. A recent review reported a rAAV-CRISPR vector that can self-inactivate its Cas9 protein by encoding an anti-Cas9 gRNA on the same construct that harbors the Cas9 itself [148]. Nevertheless, single guide RNAs (*sgRNA*) comprise short hairpin sequences that potentially cause truncation of the rAAV production similar to short-hairpin RNAs (shRNA). Placing the gRNAs close to the second ITR might increase the production yield and increase proper vector expression upon infection of the target cells [147].

### 4.4. Production and rAAV Vector Integration

#### 4.4.1. Production: The Backbones and Bacterial Resistance Genes

Impurities in rAAV products hinder the release of pharmaceutical products but might also negatively impact the potency of the expression vector. During the rAAV production, the gene therapy vector is packaged in the rAAV capsids. However, under suboptimal conditions, the capsids do also package other sequences such as vector plasmid backbones (~3%), helper plasmids (~0.05%), and even human genome sequences (~0.15%) [225]. Increasing the size of the vector plasmid backbone to above 5 kb considerably reduced inappropriate packaging and also reduced the number of empty capsids. However, larger plasmids give lower DNA plasmid yields in bacterial culture and are somewhat harder to transfect efficiently into cell lines. Jean Bennett’s group enhanced safety and maximized the therapeutic effect by adding a stuffer sequence for rAAV product hRPE65v2 [39,226]. The global health agencies (FDA, EMA, WHO) also discourage the use of β-lactams (i.e., ampicillin, penicillin) and streptomycin resistance genes in plasmids for gene therapy [227]. Kanamycin and neomycin are both members of the aminoglycoside antibiotic class. These antibiotics are tolerated, but switching to antibiotic-free systems or minimizing the use of antibiotics is preferred. Some researchers use, therefore, minicircle DNA vectors devoid of prokaryotic and antibiotic DNA sequences for their AAV production [228]. No differences were found for minicircle single-stranded rAAV2.*eGFP* production. For the self-complementary rAAV2.*GFP* vector*,* the plasmid backbone was packaged 30 times less into capsids. Moreover, the use of minicircle plasmids during rAAV production allowed for high transduction titers of rAAV vectors on HeLa cells [229].

Pro-viral plasmids can also include short-hairpin RNAs to downregulate host cell proteins that hinder rAAV capsid assembly during production. Downregulating the Y-box binding protein 1 (YB1) in HEK293T cells did increase the physical titer by 47× for rAAV2, yet it failed to improve the yield for rAAV5 [163].

#### 4.4.2. Production: ITR Stabilization

An inverted terminal repeat (ITR) of rAAV consists of an A-A’, RBE-RBE’, B-B’, C-C’, D, and terminal resolution site (*trs*) sequence (Figure 4A). The AAV Rep78 and Rep68 proteins expressed by the pHelper plasmid induce a nick on the terminal resolution site on the ITR [230]. The RBE-RBE’ initiates the binding of Rep78 and Rep68 proteins, and the B-B’ further stabilizes the proteins so the Rep78 and Rep68 can efficiently induce DNA replication during the rAAV production cycle [231]. The D sequence is the packaging signal, is important for the AAV replication, and can bind the double-stranded D sequence binding protein (ds-D-BP) [232,233,234]. When high copy numbers of a D sequence expression vector are present in HEK293 cells, then the interferon-λ-mediated activation of the major histocompatibility complex class II (MHC-II) is dampened, potentially by the binding of the ds-D-BP protein with the D-sequence instead of the X-box (RFX) regulator [235]. rAAV can be produced with only one ITR if the other ITR shows specific deletions, and ITR deletions can be recovered during production [233,236]. Moreover, replacing the 5′ ITR with a U-shaped hairpin allows rAAV production and episomal concatemerization [147]. A systematical study into ITR mutations (deleting B-B and C-C’ regions) indicated a reduced yield (4–8× fold) but a 4-fold increased transgene expression in HEK293 cells 72 hours post-infection [237]. The deletion of the BC region caused an ITR change from a T-shape to a U-shape hairpin (similar to short hairpin RNA). The authors postulated that the 2 × 34 bp ITR deletions might allow larger packaging of rAAV gene cassettes.

Recombinant AAV plasmids can disrupt ITRs in many *E. coli* cells during plasmid production or rAAV production. We also noticed an almost complete loss of one ITR within one production cycle of a pro-viral plasmid in bacterial GeneHog cells (Invitrogen) (TMB and JW, unpublished data). We validated the loss in the pro-viral plasmids by restriction enzyme digestion with XmaI at the C-C’, BssHII for the RBE, and Eam1105I for the RBE’ region. ITR Sanger sequencing has been intricate on circular DNA. We sequenced the whole ITR by first linearizing the plasmid by Eam1105I digestion at the RBE’, and then Sanger sequenced the pro-viral plasmid from both directions. The method allows us to use similar functional ITR ratios between batches (Figure 4A).

#### 4.4.3. rAAV Vector Integration into the Host Genome and Chromatin Association

Integration of rAAV into the host cell genome is unwanted because it might be genotoxic and lead to oncogenesis, especially in dividing cells and non-target cells. The rAAV gene cassettes only harbor the palindromic inverted terminal repeats (ITR) of the original wild-type AAV. All other AAV wild-type sequences are lost during the rAAV production. The ITRs are part of the Long Terminal Repeats (LTRs) family. The LTRs are extensively exploited by retrotransposons or the pro-viral DNA of retroviral RNA. ITRs, similar to LTRs, are essential to allow AAV genome integration or episomal concatemerization. The ITR hairpin structures allow self-priming (primase independent synthesis of double-stranded DNA). The ITR-gene cassette stays as a monomeric episomal form in the nucleus at low multiplicity-of-infection (MOI). At high MOIs, the ITRs form head-to-tail end-to-end joining, essentially making circularized DNA (>12kb). Further, the 5′LTR generally has a promoter function, and the 3′LTR can act as a termination sequence. Each of the ITRs of the AAV serotype 2 (ITR2) is only 145 bp long and lacks the promoter and termination functions (Figure 2B).

Chromatin-modifying proteins have been extensively studied but are viewed as less relevant for rAAV vectors because rAAV vector DNA, resides mainly in a concatemeric-episomal conformation in the nucleus, therefore potentially less regulated by epigenetics [23,239]. However, chromatin immunoprecipitation did pull down rAAV concatemer vectors after treatment with histone deacetylase inhibitor FR901228 of rAAV infected cells suggesting that rAAV might interact with histone-associated chromatin [240]. Moreover, viral DNA in minichromosomal confirmation has an identical density to chromosomal DNA indicative that chromatin-modifying factors might play a role for rAAV vectors [241]. Besides, the knockdown of chromatin assembly factor 1 increased rAAV transduction in HeLa cells [242]. Further studies need to address what kind of roles the chromatin-modifying factors have on naked DNA rAAV vectors in minichromosomal conformations in the nucleus. It is also interesting to point out that part of the infectious rAAV viral particles may redistribute to neighboring cells and may remain long term in the cytoplasm and nucleus [243].

rAAVs lack viral proteins for efficient genome integration. Integration of foreign DNA (rAAV gene cassettes) into the mammalian genome is related to the amount of double-stranded DNA (dsDNA) breaks and the DNA repair pathway that is active in the cell. Integration events can be increased by increasing dsDNA breaks in vitro by adding intron-encoded endonuclease I-SceI, etoposide, or γ-irradiation [244]. Dividing cells favor the homologous recombination (HR) DNA repair pathway during the cell cycle S-phase that requires a DNA template to guide repair such as a viral gene cassette. However, quiescent cells, such as retinal and RPE cells, favor the nonhomologous end-joining (NHEJ) pathway ligating the ends directly without the insertion of a template.

The safety profile of the rAAV relies on that upon intravenous injection, more than 85–95% of rAAV vector genomes remain episomal in the dividing hepatocytes in the mouse liver (Figure 4B; [245]). The study might have overestimated integration events in hepatocytes. Others estimate the AAV integration events closer to 0.1–1% [246]. Yet, follow-up studies indicated that 53–62% of rAAV integrations in the liver fused into actively-transcribed genes, and 3–8% into ribosomal DNA [247]. rAAV genome integration into mouse muscle tissue DNA compared to hepatocyte DNA was hard to detect or not present, indicating that the integration frequency also depends on the cell type [23]. Integration of wild-type adeno-associated virus (wtAAV) compared to rAAV for human cardiomyocytes at high MOI (50,000 viral particles per cell) was 5.6× higher with both AAVs integrating into mitochondrial DNA [248]. A recent study looked at integration events in nonhuman primates and patient DNA in clinical trials (liver biopsy) that received the rAAV2/5-*cohPBGD* and found 10^−3^ to 10^−5^ integration events per cell or 0.04–9% integration events [249]. Very little information is available for rAAV vector genome integration events into retinal tissue. A recent CRISPR/Cas9 study indicated that Cas9 breaks caused >1–20% insertion events of the rAAV cassette (EDIT-101) into the dsDNA break in the CEP290 intron in human retinal explant DNA, not counting integrations in other regions of the genome [250]. The insertion of the rAAV was higher when more indels, deletions, and inversions were detected (over 25 independent samples). The results indicate, as expected, that rAAV integrations events in photoreceptor cells correlate to the rAAV dosage. rAAV integration studies in the retina is an underrepresented research field. Almost no rAAV study specifically investigated rAAV integration events.

The integration of rAAV vectors at ribosomal DNA (rDNA) can be exploited by adding 1 kb homology arms of the rDNA locus adjacent to the ITRs. The homology arms increased the integration frequency in dividing cells favoring homologous recombination by 10–30× from a baseline of 0.001–5% AAV vector integrations per 100 cells (depending on the vector dose). Further, many rAAV gene cassette integrations caused deletions in the genome [251].

The unique T-shape AAV serotype 2 ITR-DNA (ITR2) conformation enables even gene editing. A defective eGFP reporter plasmid in dividing cells was rescued (gene-edited) by adding only the ITR2-*eGFP*part(165 bp)-ITR2 with 40 bp *eGFP* homology arms adjacent to the ITR2 [252]. Nevertheless, the rAAV gene expression from integration events is generally silenced within eight passages. Moreover, wild-type AAV integrates preferentially at 94% at the AAVS1 locus on chromosome 19 (Chr19) because the ITR sequence is homologous to the AAVS1 locus, but this requires the AAV integrases Rep78 and Rep68 that have been removed in rAAVs. Thus, rAAVs do not integrate at the AAVS1 locus. Interestingly, very little to no rAAV integration has been found in the genome of CRISPR/Cas9 gene-edited quiescent cells suggesting that off-target editing requires cell division. Unmistakably, many successful AAV gene therapies in mice and >130 rAAV clinical trials in humans, have indirectly suggested that the genome integration/genotoxic events are of a lesser concern [249].

### 4.5. Codon Optimization and Self-Complementary rAAVs

The final rAAV gene cassette could be codon-optimized to improve the optimal expression of the transgene in the target cell and organism. Essentially, codon-optimization is primarily looking at codon frequencies that might rate limit the transcription of the gene in the target cell, such as favoring the codon GUG over GUU for valine in humans. The codon optimization of rAAVs should prevent the inclusion of potential hairpin structures, repeats, extreme GC content, alternative open reading frames (ORF), and cryptic splice sites. Different codon optimizations have been tested for the rAAV gene therapy for Crigler-Najjar syndrome that increased the *FIX* transgene expression by 4–10-fold [253], and codon-optimization of rAAV2/5.hRPE65.*hRPE65* to rAAV2/5.OPTI*RPE65* might allow for higher levels of transgene expression and the use of lower rAAV titers to reduce temporal capsid-mediated toxicity [36]. For the rAAV vector AGTC-501 expressing the human *RPGR-ORF15* gene, the Codon Adaptation Index (CAI) of human codon-usage frequency was increased from 0.73 to 0.87, and the Frequency of Optimal Codons (Fop) was increased from 32% to 57%. Further, the GC content was increased, and the maximum repeat size decreased. Such adaptations in the AGTC-501 vector resulted in reduced frequency of alternative splicing and increased mRNA stability [43,110]. Interestingly, the same stabilized RPGR sequence of the AGTC-501 produced a full-length RPGR-and a truncated form of RPGR-ORF15 in the retina of mice in vivo. However, when applied to HEK293T cells in vitro, then the AGTC-501 vector produced only the full-length RPGR-ORF15 protein. The different products produced from the same *RPGR* expression cassette indicated species differences in the regulation of gene transcription or RNA splicing or differences between in vitro versus in vivo transgene expression systems. Transgene expression cassettes for clinical retina studies should, therefore, be tested in cultured human cells and preferentially in human retinal organoids or human RPE cells.

Recombinant self-complementary AAV (scAAV) vectors are more potent to express high levels of transgenes than recombinant single-stranded AAV (ssAAV) vectors. Once the cell is infected by the scAAV and the scAAV becomes decapsidated, the rate-limiting step to create double-stranded DNA is overcome more efficiently in scAAV than in single-stranded rAAV. A major disadvantage of scAAV is however the reduced packaging capacity of the gene expression cassette from up to 4.9 kb to a maximum of 2.5 kb, including the two ITRs. ScAAV mediated transgene expression could be enhanced by 5 up to 140 fold in vitro [254], and a further 90% increase in the yield of dimeric genomes was be achieved by deletion of the terminal resolution site sequence from one ITR (ITR2Δ) [255]. When compared to ssDNA, lowered rAAV doses of scAAV gene therapy vector can be used to reach similar transgene expression levels in the retina in vivo [101,152,256,257]. ScAAVs can also have significant effects on the promoter choice. The rescue by subretinal injection of scAAV8-sRLBP1p.*RLBP1* rescued the rate of dark adaptation measured by electroretinography (ERG) in *Rlbp1* knockout mice [101,152]. The transcription of *hRLBP1* by a short RLBP1 promoter increased 50-fold at a low dose (1 × 10^8^ viral genome) and 6.4 fold at a high dose (1 × 10^9^ viral genome) in cynomolgus monkey retina. The scAAV might be especially beneficial when weak promoters are required or if low viral doses are desired to prevent capsid toxicity of specific target cells. Specific details on scAAV can be found in a recent review by McCarty [255].

## 5. Transgene and Bioactivity Assays in Ocular Tissue

Quality control is essential in rAAV production. Quality control includes testing the rAAV production on safety (sterility, viral contaminants, mycoplasma, endotoxins, bacterial and fungistatic activity), appearance, pH, osmolarity, potency (viral genome titer, infectivity, expression), purity, and vector genome identity [258]. Here, we focus on the rAAV potency: infectivity and in vitro and in vivo expression.

### 5.1. In Vitro Immortalized Epithelial Cell Lines for Transgene and Bioactivity Assays

The most straight forward, high-throughput, fast, cheap, robust, but less predictive transgene expression assays are still monoculture systems in vitro. The ideal system would allow a fast characterization of the transduction profile, including rAAV capsid-specific infection, promoter expression profiling (mRNA level), and protein-of-interest expression. Such an assay would allow fast screening of different gene cassettes in research development but also validating the gene therapy products and batches in the clinical-grade production cycle.

rAAV transfection efficiency (potency: infectivity) and transgene expression (potency: in vitro expression) has been tested on rAAV epithelial production cell lines (for example: HEK293T, HER911, HeLa-E1, Per.C6). One can measure the rAAV infection or functional titers measured in transducing units per mL (TU/mL) by two common assays: (1) median tissue culture infective dose (TCID50) based on rAAV-vector (MOI 20,000 viral genome/cell) infecting HeLaRC32 (HeLa AAV rep-cap expressing cell line) and concomitant adenovirus type 5 (Ad5; 500 IUs/cell) infection (Outcome: vector genome quantification). (2) Infectious center assay (ICA) based on HeLaRC32 and Ad5 (500 IUs/cell) infection (Outcome measure: hybridization of a probe to the rAAV gene cassette). The HeLaRC32 cells allow rAAV replication if the rAAV gene cassette can enter the nucleus [259].

Spark Therapeutics also developed an in vitro potency assay for rAAV-*RPE65* vectors. Here, modified HEK293 cells constitutively expressing lecithin-retinol acyltransferase (LRAT) are transduced by the rAAV-*RPE65* at different MOIs. Then, 72 hours later, the cells are lysed by adding lysis buffer, and the lysate is incubated with all-trans-retinol and CRALBP for 2 hours in the dark to assess the enzymatic activity of RPE65 (an isomerohydrolase) to convert all-trans-retinol to 11-cis-retinol [260]. The rAAV-*REP1* for the treatment of choroideremia is assessed on in vitro prenylation of RAB6A in HEK293 cells [261]. The directly injectable dose can also be assayed. The potency of the residual diluted vector (rAAV2-*REP1*) from the syringe used to inject patients was applied to the REP1-deficient cell line HT1080 for REP1 expression [155]. An endothelial-like Human Trabecular Meshwork (HTM) immortalized cell line is currently explored for glaucoma rAAV gene therapy. Still, the surrogate cell line might also be interesting for studying the off-target effects of rAAV vectors injected intravitreally [262,263]. However, it is not always possible to develop informative assays on epithelial cells, especially for large screens for novel retinal-specific capsids or retinal specific promoters. For example, rAAV5 vectors infect very poorly HEK293 cells, HeLa cells, and BJ fibroblasts [264] but infect RPE cells and human retinal organoids efficiently [62,265]. Consequently, if a rAAV5-CAG vector does not express a transgene product in a HEK293 assay, then the vector might still express the transgene in human RPE or human retinal organoids.

### 5.2. In Vitro Immortalized Ocular Cell Lines for Transgene and Bioactivity Assays

Researchers can also use various human retinal cell lines to achieve improved infection and expression of retinal specific capsids or promoters. Promising ocular cell lines are the 661W mouse photoreceptor cell line/retinal ganglion precursor-like cell line (a surrogate for cone photoreceptors), the Adult Retinal Pigment Epithelial cell line-19 (ARPE-19; a surrogate for RPE cells), hTERT RPE-1 (ATCC^®^ CRL-4000^™^), human astrocytes and the MIO-M1 (surrogates for Müller glial cells).

661W mouse cell line do not express GFAP but do express neuronal genes such as *Opn1SW*, *Opn1mw*, *Rbpms*, *Brn3b*, *Brn3c*, *Thy1*, γ-synuclein, nestin, *NeuN*, *Map2C*, *Map2D*, and β-III tubulin. The 661W cells are light-sensitive but do not have visible outer segments [266,267]. 661W and ARPE-19 cells allow for screening of RPE-specific (mCARpro, MOPS500, VMD2) and ubiquitous (smCBA) promoters for most rAAV serotypes [268]. The 661W cell line has been further modified to achieve improved transfection by overexpression of the universal adeno-associated virus receptor (AAVR) or more stably express key photoreceptor genes such as *GRK1* and *CAR* by rAAV-VPR-*dCas9* vector infection [269]. Two major drawbacks to immortalized ocular cell lines are that most tend to be very heterogeneous, some of the cell lines express multiple cell-type-specific markers such as for retinal ganglion cells (Brn3) as well as for cones (Opn1mw) in addition to neuronal cell markers (Nestin, NeuN). Moreover, none of the cell lines have mature photoreceptor outer segments. Another RPE cell line (hTERT RPE1; ATCC^®^ CRL-4000^™^) was used for liposome co-flotation assays expressing a biological active truncated CEP290 (1-580 amino acid) protein (rAAV-*CEP290^1-580aa^*) in the primary cilium [270]. A *CEP290* knockout hTERT RPE1 line was constructed, showing the cilia-related CEP290-phenotype [271].

Human primary astrocytes have been successfully employed to select for novel rAAV capsid variants that are specific for (Müller) glial cells. An example are the rAAV6 variants ShH10 and ShH10-Y445F that efficiently infect rat, mouse, and human Müller glial cells [60,62,272]. Human astrocytes also express common Müller glial cell markers such as SOX9, GFAP, GLAST, GS, Kir4.1, and S100β [273,274]. A human Müller glial-like cell line (MIO-M1) expresses the proteins GLUL, VIM, low GFAP (but found on mRNA level), RLBP1, GLAST, EGFR, SLCA1, AQP4, Kir4.1, THY1, NEFH, MAP2, NEUROD1, NEUN, Nestin, SOX2, Chx10, PAX6, NOTCH1, βIII tubulin. However, the Müller glial-like cell line also contains mRNA for the following opsins or visual cycle-related genes: *OPN1SW*, *OPN2*, *OPN3*, *OPN4*, *OPN5*, *RRH*, *GNAZ*, *GNAT1*, and *GNAT2* [275,276]. The MIO-M1 cell line has been successfully used to screen for rAAV infectivity, rAAV, and lentivirus cell-specific promoter expression [277]. The hypoxia-Müller glial specific promoter (scAAV2.HRSE.6xHRE.GfaABC1D.*luciferase*) is active in MIO-M1 cells under hypoxic conditions. The hypoxia-induced Müller glial specific promoter showed no luciferase expression in HEK293, C6, HT22, and ARPE19 cells [97].

All described cell lines hold great promise for further rAAV studies. One needs to be cautious of the results because (1) changes in culture condition can strongly affect the “cell-specific” gene expression, (2) cell contamination has been found in several lines such as the rat ganglion cell line 5 (RGC-5) being a subclone of the 661W cell line, (3) multiple cell-type-specific gene markers expressed for example in 661W, and (4) the overall lower biologically relevance compared to 2D and 3D cell or in vivo studies.

### 5.3. In Vitro Differentiation of Human Induced Pluripotent Stem Cells (hiPSCs) to Retinal Pigment Epithelium (RPE) Cells

Human patient induced pluripotent stem cells (hiPSCs) can be differentiated to photoreceptors [278]. 2D differentiation of hiPSCs to photoreceptors peaks at 45 days of differentiation, but it declines fast, making rAAV studies difficult because of the short time window and the inherent instability of inner/outer segments.

Human patient induced pluripotent stem cells (hiPSCs) can also be differentiated to monolayers of RPE. rAAV-*REP1* vector transduction of patient hiPSC-derived *CHM*-RPE rescued the biochemical phenotype [279]. The patient hiPSC-derived RPE can be efficiently used for testing the AAV-*CHM* vector for rescuing prenylation, phagocytosis, and protein trafficking [280]. Moreover, a proof-of-concept for dominant retinitis pigmentosa due to haploinsufficiency rescued phagocytosis and cilia formation by AAV2/Anc80-*PRPF31* in hiPSC-derived *PRPF31^+/−^* RPE cells [281]. Spark Therapeutics filed a patent application for a potency assay of rAAV-*CHM* on hiPSC-derived RPE cells lacking *CHM* expression [282].

### 5.4. In Vitro Differentiation of Human Induced Pluripotent Stem Cells (hiPSCs) to Retinal Organoids for Transgene and Bioactivity Assays

We and other researchers use human retinal organoids to study rAAV transduction and potency [62,265,283,284]. In summary, photoreceptors are transduced by rAAVs such as rAAV2, rAAV2-7m8, rAAV5, rShH10, rShH10-Y445F, rAAV8, rAAV8T(Y733F), and rAAV9 albeit at different transduction efficacies. The rAAV2-7m8, rAAV5, rShH10, and rShH10Y-445F capsids infect photoreceptors efficiently. Interestingly though, (early) radial retinal progenitor cells in retinal organoids or common cell lines can efficiently be infected by rAAV6, the rAAV6 variants (ShH10 and ShH10Y-445F), and the AAV2-7m8 [21,62,265,284]. For example, rAAV6 and rAAV-derived vectors (ShH10; ShH10-Y445F) infected hiPSCs and hiPSC-derived RPE cells properly and thoroughly [21,62,265]. rAAV2-7m8 and rAAV5 also efficiently infected RPE cells [265,284].

Many challenges still lie ahead of us. We need to learn to properly control rAAV infection of different rAAV capsid compositions. The presence of rAAV receptor-mediated endocytosis is influenced by the docking on of capsid proteins to the cellular receptors on the cell surface. Medium compositions alter the extracellular matrix proteins and has influence on rAAV endocytosis. For example, fibroblast growth factor receptors (FGFRs) are important for the stabilization of the heparan sulfate proteoglycans (HSPGs) on the extracellular matrix of the cell [285]. FGF-2 binds with low affinity to the heparin sulfate chains of HSPGs and the FGFRs [286]. This growth factor can be found in many medium compositions as a supplement or in fetal bovine serum (FBS; 8–45 pg/mL) [287]. Many AAV capsids require the HSPGs, the universal AAV receptor (AAVR), and the FGFRs for efficient cell entry. For example, rAAV2 requires the FGFR1 receptor that can be blocked by FGF-2 supplementation [288]. Other rAAV serotypes, such as AAV4, are less dependent on the HSPGs and FGFRs for rAAV capsid cell entry [285]. Thus, the FGF-2 concentration in the medium needs to be defined for rAAV potency assays. Not all co-receptors for rAAV entries have been discovered yet, adding to the uncertainty of rAAV potency assay data.

Other uncertainties related to the biological relevance and the transgene and bioactivity assays with retinal organoids are: the loss of ganglion cells in long-term culture, improper lamination of the ganglion cell layer (including astrocytes), no innervation of the optic nerve (required for proper foveal development), no vascularization of retinal organoids (therefore, no pericytes), no sclera and Bruch’s-membrane (blood–brain barrier), no immune cells (macrophages, microglia, dendritic cells), and no integration with other organs (brain, heart, liver, kidney). Another uncertainty is that the quality of the starting material (hiPSCs) and the culturing protocol can significantly affect the success in differentiation to retinal organoids inadvertently affecting rAAV tropism [289,290]. More sophisticated models are currently in development. For example, human iPSC-derived retinal organoids and RPE-sheets can be cultured in a microfluidic chip system that enhances photoreceptor maturation and stabilization in vitro [291].

### 5.5. Human Ex Vivo Retinal Culture for Transgene and Bioactivity Assays

We and others have also demonstrated that rAAVs can be tested on ex vivo cadaveric human retinas [50,62,250,292,293,294,295,296]. Ex vivo studies have especially become more attractive since the advent of more efficient medium compositions with better inner-outer photoreceptor segment (IS/OS) quality and ganglion cell survival that allow for longer rAAV-transgene expression. rAAV1, rAAV2[MAX], rAAV2(quad Y-F), rAAV2-7m8, rAAV4, rAAV5, rAAV6 and to a lesser degree rAAV2 and rAAV9 can efficiently infect photoreceptors on ex vivo human cadaveric retinas. We have shown that rAAV5 and the rAAV6 variants ShH10 and ShH10-Y445F can efficiently infect both human photoreceptors and Müller glial cells ex vivo [60,62]. rAAV2(quad Y-F) and rAAV2-7m8 also infected Müller glial cells and some rod photoreceptor cells ex vivo [293,294]. rAAV2/8BP2 infected photoreceptors, Müller glial cells, amacrine cells, ganglion cells, and horizontal cells ex vivo [50] Moreover, the hRLBP1 promoter (2.6 kb) can restrict expression to Müller glial cells and RPE [60]. Recently, rAAV5.GRK1.*SaCas9* vector particles were added to retinal explants that were subsequently cultured for 28 days. The human photoreceptor-specific rhodopsin kinase (GRK1) promoter directed Cas9 expression specifically in human photoreceptors [250]. We have also shown that the GRK1 promoter can limit expression to photoreceptor cells in ex vivo cadaveric human retinas [297].

We have previously shown that mouse retinal explants at P0 can be cultured longer than adult retinal explants [189]. Likewise, human fetal retinae can be cultured for three weeks, with preservation of the general morphology preserved and only the loss of ganglion cells [298]. More astonishingly, fetal retinal tissue that maintained some photoreceptor morphology has been cultured for 293 days in vitro [298]. However, the access to and quality of the donor human fetal material limits the application for many researchers [62,299,300]. We observed that efficient transduction of photoreceptors by rAAVs required the presence of well-developed inner/outer segments of photoreceptors. Interestingly, rAAVs infected Müller glial cells more efficiently in explants that showed retinal degeneration [62,294].

### 5.6. In Vivo Studies for Transgene and Bioactivity Assays

#### 5.6.1. Developmental Stage and rAAV Infection

Many adult mouse rAAV potency studies have been performed and are reviewed elsewhere [2,7]. It is interesting to note that rAAV infection can differ depending on the developmental stage of the ocular tissue. Mouse fetal retina (embryonic day 13; subretinal injection) can be transduced well by rAAV5 but not by rAAV1 and rAAV2. However, rAAV1, rAAV2, and rAAV5 transduce photoreceptors well at P30 [300]. Mouse photoreceptor cells at postnatal day 0 can be efficiently targeted by subretinal injection of rAAV1, rAAV5, rAAV9, and rAAV11 [122]. However, AAV1 transduces mainly RPE cells at the adult stage [2,7]. Moreover, rAAV8-CMV-*GFP* transduces photoreceptors and Müller glial cells at postnatal day six but only transduces photoreceptors at postnatal day 0 [301]. Here, the rAAV vector containing the CAG promoter compared to the CMV promoter showed expression in more cells (photoreceptors, horizontal cells, amacrine cells, and retinal ganglion cells vs. only photoreceptors) at postnatal day 0. The difference in the infection efficiency at early development compared to more mature retinas has been linked to rAAV receptors and co-receptors important for rAAV cell entry [284]. The main universal AAV receptor (AAVR) was present already in 44-day-old human retinal organoids. However, the poor transfection efficiency of rAAV9 of retinal organoids [62,265] was linked to the low abundance of N-linked-galactose at early retinal developmental stages [284]. It points to the importance of describing the receptor composition and medium composition accurately at the time of infection to make meaningful comparisons between the infectivity of different rAAV capsids.

#### 5.6.2. rAAVs Overcoming Membranes in the Retina and the Retinal Disease State

Delivery of rAAV gene supplementation to photoreceptors or RPE cells is generally done by subretinal injection because many rAAVs cannot penetrate through the inner limiting membrane (ILM) when injected intravitreally. The ILM is close to the ganglion cells and the Müller glial endfeet with a thickness varying between 100 nm up to 2000 nm in nonhuman primates [302]. The rAAV subretinal injection creates a fluid bleb between the retina and the RPE layer causing temporal retinal detachment and infection of cells at foci. Intravitreal injections target a larger retinal area. Intravitreal injections of empty rAAV capsids can induce a temporary immune inflammation of the aqueous and the vitreous [303]. Enzymatic digestion (proteasome inhibitors) of the ILM or ILM/outer limiting membrane (OLM) or disruptions of the ILM/OLM by the disease can alter rAAV infection and allow rAAV infection deeper into the retina [99,294,304,305]. Applying a low trans-ocular electric current also allowed efficient transduction of RPE and photoreceptors by rAAV8 upon intravitreal injection in adult mice [306]. Finally, the application of tyrosine kinase inhibitors might improve the passage of rAAV through the ILM or OLM [307]. The novel methods might make the intravitreal injection more common for photoreceptor and Müller glial cell infection. Intravitreal injection of some rAAVs might result in the transduction of a larger pool of off-target cells, such as the ciliary body and iris epithelium [60].

Disease-induced changes to the retinal morphology do impact the rAAV infectivity. In many retinal diseases, for example in *Crb1* retinitis pigmentosa mouse models, we first find that Müller glial cells express stress markers (gliosis), the outer limiting membrane (OLM), which contains adherens junctions between photoreceptors and Müller glial cells disrupts at foci, the inner/outer segments of photoreceptors shrink, some of the photoreceptors die, macrophages and microglial cells are activated and assemble in the photoreceptor segment layers, the outer and inner nuclear layers mix and thin out, and neovascularization takes place [62,189,190,191,192,193,194,195,196,197,198,199,200,201,202]. Likewise, transgenic rats overexpressing rhodopsin variants causing autosomal dominant retinitis pigmentosa show an early (P20), intermediate (P30), and advanced stages of retinal degeneration (P60). rAAV1 or rAAV5 intravitreally injected in rats showed no accumulation of AAV particles at the ILM [304]. However, under the disease condition or enzymatic digestion of the ILM, all rAAV (1, 2, 5, 8, and 9) traversed, most likely via Müller glial cells, to RPE cells through the retina [99,304,308]. rAAV2-7m8 vectors infected a wide range of cells in the degenerate retina of rd1 mice (*Pde6b*^rd1/rd1^) with little differences found in tropism when the vector was injected subretinally or intravitreally [294]. The potency to infect the degenerate photoreceptors by subretinal injection of *Abca4*^KO^ mice compared to wild-type mice was lower for four different rAAVs (rAAV2, rAAV5, rAAV2rec2, rAAV2rec3) while the potency to infect INL cells was increased in *Pde6b*^rd1/rd1^ mice at least for rAAV5, rAAV2/Rec2 [309]. Intravitreal injection of rAAVrh-10 shifted infection from mainly INL cells towards photoreceptor and RPE cells in *Rs1*^KO^ and *Rho*^KO^ mice (XLRS and RP models, respectively) compared to wild-type mice [310,311]. The studies indicate that disease models allow deeper penetration of rAAVs in disease state compared to healthy retinas with generally lower photoreceptor infection. How well changes in the OLM and ILM impacts rAAV infection needs to be determined separately for each type of retinal disease (Figure 5).

The changes in rAAV vector transduction and expression can be linked to the differentiation stage of cells and cellular stress. Non-dividing terminally differentiated cells allow efficient expression from rAAV gene therapy vectors because the cells downregulate proteins of the DNA damage response [312]. In AMD and retinitis pigmentosa, retinal cells might show an increase in DNA damage response, including more double-stranded DNA breaks and impaired (decreased) autophagy leading to increased cell size, granularity, and protein accumulation [313,314,315]. How the increase in DNA-damage sensors or the decrease of terminal differentiation is linked to rAAV vector transgene expression needs to be further evaluated. Cell stress-induced expression was observed upon induction of gliosis, by light or application of ciliary neurotrophic factor (*CNTF*), in *Crb1*-deficient retina injected intravitreally with rAAV.GFAP.*eGFP* [99].

#### 5.6.3. Nonhuman Primate Studies and rAAV Infection

Several rAAV serotypes are studied for safety/toxicological assessment in retinas of nonhuman primates (NHP) before entering clinical trials. Some information on tropism and promoter cell specificity has been acquired. Subretinal injection (ubiquitous promoter): rAAV1, rAAVrh64R1, rAAV2, rAAV2-7m8, rAAV5, rAAVrh8R, rAAV7, rAAV8, rAAV8BP2, rAAV9, rAnc80L65 infect rod photoreceptors and RPE cells [280,293,316,317,318,319,320]. rAAV8BP2, rAAV9 and rAAV5 infect cone photoreceptors more efficiently than rAAV2 [318,319,321]. Novel AAV capsids, such as the rAAV7m8 and rAAV8BP2, also infected some INL cells and ganglion cells [50,319]. High titer rAnc80L65 and rAAV5 might infect some NHP Müller glial cells [320]. Intravitreal injection (ubiquitous promoter): rAAV2 infects Müller glial cells and ganglion cells [322]. rAAV2-7m8 infects well Müller glial cells and retinal ganglion cells, whereas rAAV8BP2 infects ganglion cells. Self-complementary rAAV2tYFinfects Müller glial cells, at least if the ILM is peeled off before injection [323]. rAAVrh-10 showed promise in transducing the whole rabbit retina and patches of RPE cells, including photoreceptors [310].

Different cell-type-specific promoters have been tested by subretinal or intravitreal injection in NHPs with surprising results. The hGRK1 promoter expressed GFP specifically in rod- and cone photoreceptors [321]. The strong cone-specific promoters in mice (mCAR, PR2.1, and PR1.7) all showed some rod expression and strong cone expression. However, the mouse cone arrestin promoter (mCAR) also expressed eGFP in rods, inner nuclear layer cells, and ganglion cells [293]. A large scale study compared novel synthetic cell-specific promoters using previously described rAAV serotypes (AAV8, AAV9, AAV8BP2) between mice, NHPs, and human retinal explants [50]. Unsurprisingly, NHP and human retinal explants matched closer (correlation r = 0.66–0.67) compared to mouse/NHP or mouse/human (r = 0.34–0.38; r = 0.24–0.32). However, a predictive value mean correlation of 0.3 demands that rAAVs and cell-specific promoters need to be tested in human systems. Nevertheless, if an rAAV vector expressed well in human retinal explants and NHP, then it is more likely (conditional probability = 0.12–0.14) that the rAAV vector works as well in the mouse retina. The results suggest, therefore, that mouse retina can be used to pre-screen rAAV vectors for cell-type specificity.

#### 5.6.4. Cis-Regulatory Toxicity of rAAV Vectors In Vivo?

AAV retinal-specific promoters were compared by subretinal injection at P0 for cis-regulatory sequence toxicity in the CD-1 albino mouse line [324]. This CD-1 mouse line is prone to hearing and vision loss, whereas albino mice are more susceptible to light-induced retinal damage [325]. The encapsidated rAAV vectors containing cis-regulatory ubiquitous promoters (CMV, CAG) or the RPE-specific promoter (hBEST1) showed higher dose-dependent toxicity to the RPE and photoreceptors than photoreceptor-specific promoters (Rho; RedO, CAR, GRK1). The ubiquitous UbC promoter showed no toxicity to the mouse RPE [324]. Interestingly, C57BL/6J mouse retinas injected with AAV8- or AAV5-CMV-*GFP* at P0 and subsequently analyzed at P30 showed RPE aberrations by SD-OCT but no ERG or OKT differences at 3 × 10^9^ vector genome copies [324]. However, stronger rAAV vectors such as the ones that contain a CMV promoter and a WPRE element such as the vector rAAV2/8.CMV.*eGFP*.WPRE.bGHpolyA caused retinal degeneration in mice at 5 × 10^10^ vg (ONL reduction and ERG) [326].

Interestingly, photoreceptor cells that were infected by rAAVs carrying a non-coding gene cassette (rAAV-flox vector) that is floxed out in Cre recombinase expressing cells also caused toxicity at 1 × 10^11^ vg or above, indicating that the rAAV-capsid can cause toxicity [326]. Further, the ubiquitous CAG promoter caused more toxicity than the rhodopsin specific promoter. Thus, further studies on how promoters might activate the innate immune system by TLR2 or TLR-9 activation, what sequence motifs are more prone to induce toxicity, or what time points are especially sensitive are of great importance. However, antigen-presenting cells can take up capsids and express the antigens on MHC class II receptors that can activate CD4^+^ T-helper cells releasing cytokines that stimulate CD8^+^ T-cells. This immune cascade model might explain why re-administration in patients could become difficult, and why the administration of rAAVs in patients with high neutralizing antibodies (nAbs) are generally not included in clinical trials [327]. However, the innate, humoral, and cell-mediated immune response might contribute to vector toxicity. For example, intravitreally injected empty rAAV capsids can induce a transient inflammation of the aqueous and the vitreous body [303]. Most importantly, cis-regulatory-sequence, rAAV-capsid, and transgene-related toxicity need to be investigated in the retinal degeneration model. Further, the more efficient gene expression cassettes might allow a rescue strategy at lower rAAV titers, thereby causing less toxicity and lower transient inflammation [303].

## 6. Concluding Remarks and Future Prospects

The tools for modulating rAAV gene therapy vectors are expanding rapidly. Novel rAAV capsids, production platforms, (short) promoters, stabilizing introns, and polyadenylation sequences are being continuously developed and made available in publications and patents. However, many papers on rAAV ocular trials do not sufficiently describe the elements on the plasmid used for the viral vector production (bacterial origin, bacterial selection markers). The vast expanse of rAAV vectors and the poor description of the production process and gene cassette elements make a fair comparison of rAAV vector elements very challenging. Large-scale comparative rAAV element studies are missing in ophthalmology but recent studies indicate that:The use of tyrosine-mutated rAAV2 capsids (AAV2-tYF; AAV2-7m8) increases retinal penetration and infection potentially replacing wild-type capsids (Section 2 and Section 5.6).The strong viral promoter CAG expresses the transgenes in the RPE for many years without being silenced [22] (Section 2, Section 3 and Section 5.6).Native promoters are more prone to differ in transgene expression levels in healthy-vs-disease states (Section 3.4 and Section 5).Promoters in general can greatly differ within in vivo/in vitro/ex vivo models, as well as across species (Section 3.4 and Section 5.6).Inducible promoters (riboswitches and dead-Cas9) offer exciting opportunities to control protein expression (Section 3.8).Surrogate gene (homolog/ortholog or synthetic) and minigene supplementation may circumvent cellular immunogenicity (Section 4.1).The rAAV production cell line and production cell line related impurities can influence the transduction efficiency in target tissue [44] (Section 2 and Section 4.4).Inverted terminal repeats of rAAVs are essential for high production yields but is not a prerequisite for the efficient transgene expression (Section 4.4.2).Genome integrations of rAAV vectors and the potential cell-toxic effect of genome integrations have been insufficiently studied in retinal tissue (Section 4.4.3).The rAAV infection pathway can vastly differ depending on the selected medium composition, the culturing technique/protocol, and the studied developmental state of the tissue (Section 5.6.1).The disease state strongly influences rAAV-vector penetration, potency, and tropism of the retina (Section 5.6.2).

A large panel of models are available for studying retinal diseases, however, all models have inherent drawbacks. We believe that improved models will become available, allowing more rapid screening of a library of rAAV promoters and rAAV capsids in human systems of high biological relevance (Figure 6, Section 5.1, Section 5.2, Section 5.3, Section 5.4, Section 5.5 and Section 5.6).

## 7. Material and Methods

### 7.1. A Meta-Analysis on Pro-Viral Plasmids and Production Platforms for Ocular rAAV Therapies in Clinical Trials

First we collected clinical trial identifiers from reviews on ocular gene therapy. Then, we searched for the keywords (AAV, gene therapy, retinitis pigmentosa, RP, Leber congenital amaurosis, LCA, AMD, CHM) on https://clinicaltrials.gov/ (last date 01 April 2020). The search was limited to ocular gene therapies. We further cross-checked the compiled list with reported clinical trials in the news as well as in current reviews. If sequences or rAAV production platforms were insufficiently reported in research papers the search was broadened to: (1) Ph.D. theses; (2) patents selected based on researchers and companies involved; (3) company registration documents (SEC filings); (4) company websites; (5) company-provided presentations; (6) results presented in international research conferences, and (7) abstracts from international research conferences. Any related information is listed in the supplementary. We matched clinical trial identifiers to each unique rAAV product. Lentiviral products, cell therapy products, antibodies, and antisense oligonucleotide (AONs) products were removed from Figure 3 and Table 1 to include only rAAV-based therapies. All the unique rAAV ocular gene therapy products were then analyzed concerning prokaryotic plasmid backbone, inverted terminal repeat AAV serotype, enhancers, promoters, introns, cDNA genes, codon-optimization, human or other species-related sequences, post-transcriptional stabilizing sequences, polyadenylation sequences, and rAAV production platform. Any missing information is indicated as “UntoldR” or by a “?”. The data was curated in MS Excel (Microsoft software) and then loaded in GraphPad Prism version 7 (GraphPad software). Figure 1, Figure 2, Figure 4 and Figure 5 were generated in MS PowerPoint (Microsoft software). All figures were exported as vector graphs and imported in Photoshop (Adobe Software) for final type-set editing.

### 7.2. Statistical Analysis

Statistical analyses and visualization for Figure 3 was done in GraphPad Prism version 7 (GraphPad software).

## Figures and Tables

**Figure 1 ijms-21-04197-f001:**
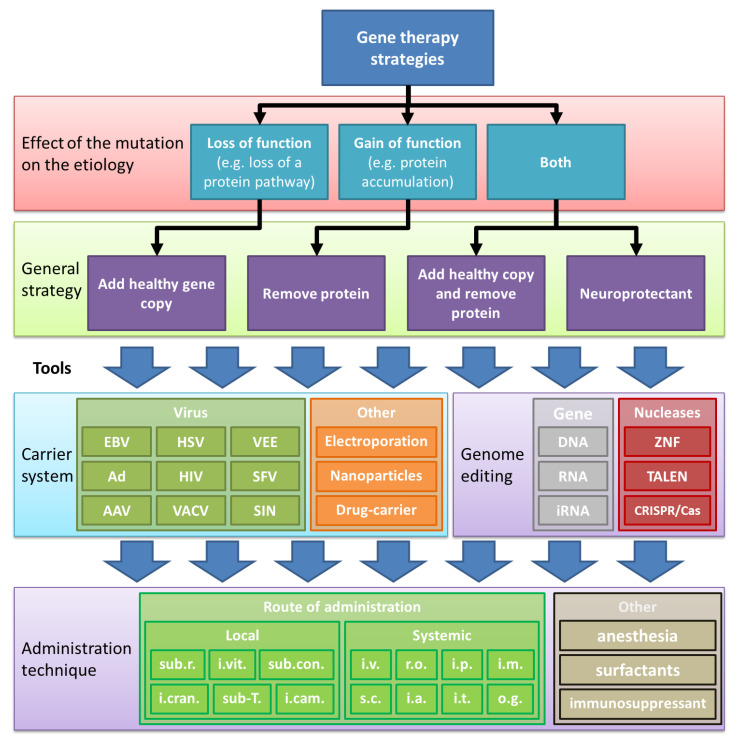
An overview of retinal gene therapy strategies. The effects of the gene variations determine the gene therapy rescue strategy to be applied. Physical DNA delivery includes electroporation, sonoporation, magnetofection, and bioballistic (gene gun) methods. Viruses: AAV, Adeno-associated virus; Ad, Adenovirus; alpha, alphavirus; Epstein-Barr virus (EBV); FV, Foamy virus; HSV, Herpes simplex virus; HIV, Human immunodeficiency virus; VACV, Vaccinia virus. Nucleases: ZNF, Zinc-finger nuclease; TALEN, transcription activator-like effector nuclease; CRISPR/Cas, clustered regulatory interspaced short palindromic repeat (CRISPR)/Cas-based RNA-guided DNA endonuclease. Route of administration: i.cranial, intracranial; i.a., intraarterial; i.cam., intracameral; i.m., intramuscular; i.v., i.t., intrathecal; intravenous (e.g., tail vein or facial vein); i.vit, intravitreal; o.g., oral gavage; r.o., retro-orbital; sub.r, subretinal; sub-T, sub-Tenon; sub.con., sub conjunctiva; s.c., subcutaneous.

**Figure 2 ijms-21-04197-f002:**
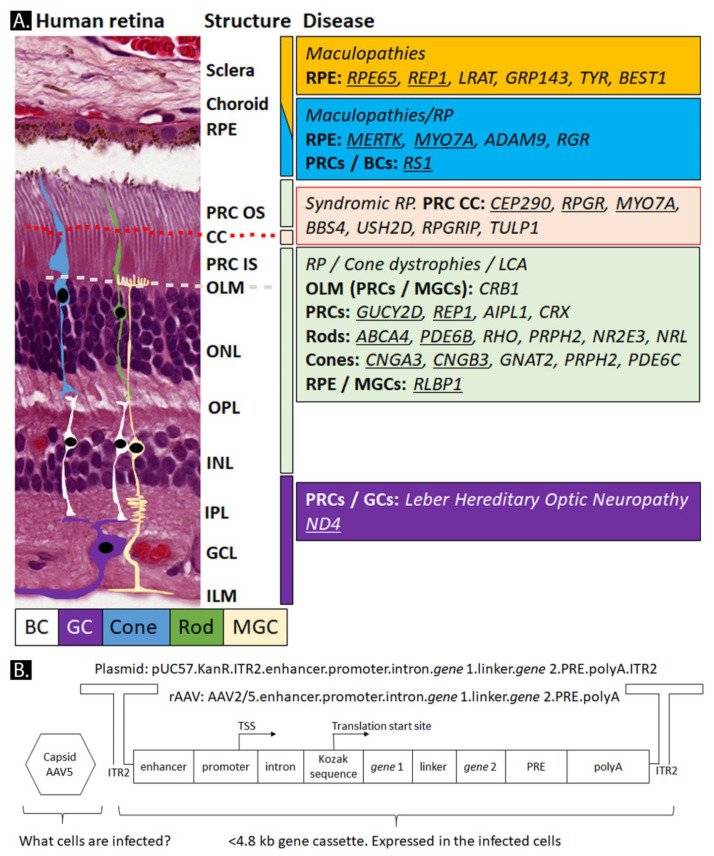
The development of recombinant AAV vectors targeting ocular diseases. (**A**) Main location (arrow) of frequent gene products (genes indicated) causing retinal diseases (color), and recombinant adeno-associated viral vectors (rAAV)-gene supplementation therapy genes (in clinical trials; underlined genes). (**B**) Hypothetical rAAV gene cassette and the corresponding plasmid. AAV5, adeno-associated viral vector serotype 5; BC, bipolar cell; Cone, cone photoreceptor; CC, connecting cilium; GC, ganglion cell; GCL, Ganglion Cell Layer; ILM, Inner Limiting Membrane; IPL, Inner Plexiform Layer; ITR, inverted terminal repeat; KanR, kanamycin resistance gene; LCA, Leber congenital amaurosis; MGC, Müller glial cell; ONL, outer nuclear layer; OPL, Outer Plexiform Layer; PRC IS, photoreceptor inner segment; PRC OS, photoreceptor outer segment; polyA, polyadenylation sequence; PRE, post-transcriptional gene regulatory element; RP, retinitis pigmentosa; RPE, retinal pigment epithelium; rod, rod photoreceptor; TSS, transcription start site.

**Figure 3 ijms-21-04197-f003:**
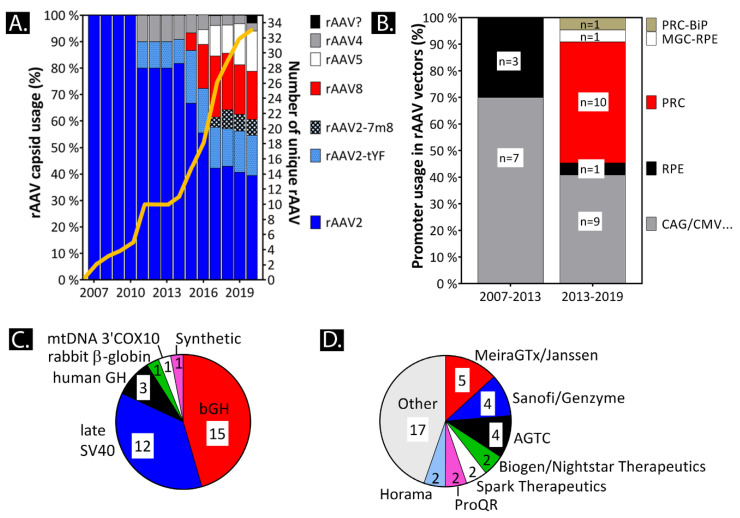
Development of rAAV therapies over the years. (**A**) Unique rAAV capsids usage over time (%; y-axis left) and unique rAAV treatments (genes) in clinical trials (number; y-axis right; Total = 33 of 57 rAAV total clinical trials for the retina). (**B**) Unique rAAV treatments (genes) and their promoters for the retina (2007–2013 vs. 2013–2019). (**C**) Unique rAAV treatments (genes) and their polyadenylation sequences in the retina (total = 28). (**D**) Sponsors with unique rAAV, Lentiviral vector, and AON treatments in clinical trials for the retina/choroid (genes; %; Total = 38). bGH, bovine growth hormone; CAG/CMV, ubiquitous promoters; late SV40, late Simian Virus; MG, Müller glial cell; RPE, retinal pigment epithelium; PRC-BiP, photoreceptor-bipolar-specific promoter; PRC, photoreceptor-specific promoter; rb β-globin, rabbit β-globin polyadenylation sequence.

**Figure 4 ijms-21-04197-f004:**
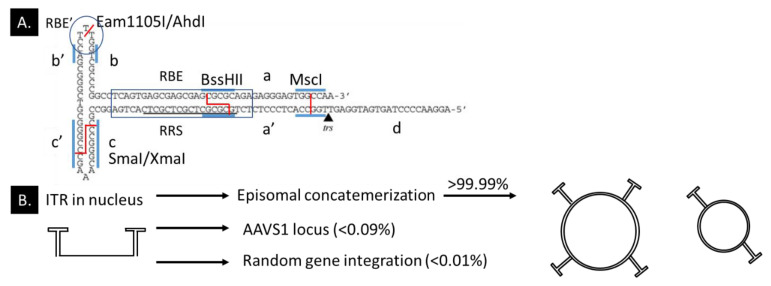
How to assess Inverted Terminal Repeats (ITRs) of rAAV and how they allow concatemerization. (**A**) Restriction enzyme sites in the AAV serotype 2 ITR in the flop configuration. RBE’/RBE binds Rep68 (RBE, Rep-binding element) and initiates the Rep helicase. The Rep helicase nicks the *trs* (terminal resolution site). Restriction enzyme recognition site indicated in blue and the actual cut in red. Figure adapted from [238]. (**B**) ITR structure in the nucleus after second-strand DNA synthesis in dividing cells favoring homologous recombination. Most rAAV-vectors form episomal concatemeric circular double-stranded DNA.

**Figure 5 ijms-21-04197-f005:**
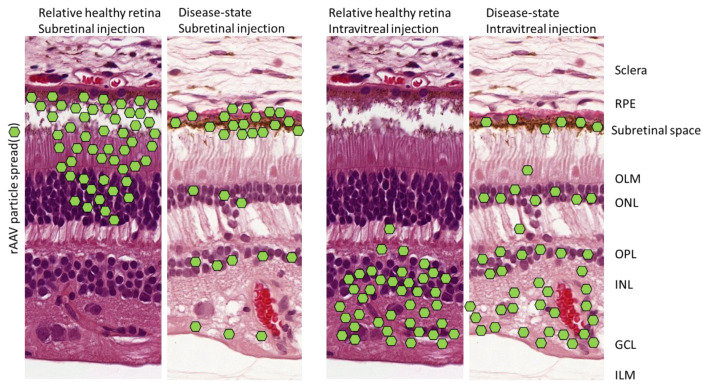
A hypothetical model of the spread of rAAV capsids (serotypes 1, 2, 5 8, and 9) after intravitreal or subretinal injection in disease or non-disease mouse retinas in vivo based on the studies [99,294,304,306,308,309]. RPE, retinal pigment epithelium; OLM, outer limiting membrane, ONL, outer nuclear layer; OPL, outer plexiform layer; INL, inner nuclear layer; GCL, ganglion cell layer; ILM, Inner Limiting Membrane.

**Figure 6 ijms-21-04197-f006:**
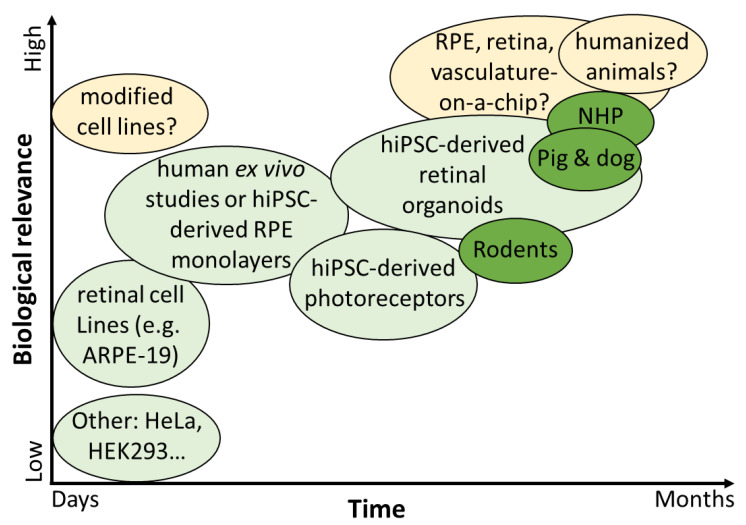
Qualitative assessment of biological relevance and time of assay for retina-specific rAAV potency assay models. Abbreviations: hiPSC, human induced pluripotent stem cell; NHP, non-human primates; RPE, retinal pigment epithelium.

**Table 1 ijms-21-04197-t001:** rAAV gene therapy products registered on clinicaltrials.gov.

Disease	Year	Product	Capsid	Promoter	Intron	Other	Gene	PolyA	Production
LCA	2007	h*RPE65*v2	AAV2	CAG	β-globin		*RPE65*	bGH	HEK293
LCA	2007	AAV-*RPE65*	AAV2	CB-SB			*RPE65*	SV40	HEK293
LCA	2008	tgAAG76	AAV2	hRPE65			*RPE65*	bGH	B50, helper adenovirus
LCA	2011	HORA-*RPE65*	AAV4	hRPE65			*RPE65*	bGH	HEK293
LCA	2016	OPTIRPE	AAV5	NA65	SV40		*RPE65*	SV40	HEK293
AMD	2009	*sFLT01*	AAV2	CAG	β-globin		*sFLT01*	bGH	HEK293
AMD	2011	OXB-201	EIAV	CMV		IRES + WPRE	*Endo+ Angio*	SIN-LTR	?
AMD	2011	AAV.*sFlt-1*	AAV2	CMV	Chimeric intron		*sFlt-1*	SV40	HEK293
AMD	2017	RGX-314	AAV8	CAG/CB7	β-globin		*aVEGFAfabH.F2A* *.aVEGFfabL*	rabbit β-globin	?
AMD	2018	HMR59	AAV2	CAG	β-globin^SD/SA^		*sCD59*	bGH	?
AMD	2018	ADVM-022	AAV2-7m8	CMV	β-globin^SD^ Ig^SA^	TLP-eMLP	*sFLT01co*		Sf9
AMD	2019	GT005	AAV2	CBA	β-globin	WPRE	*CFI*	bGH	?
LHON	2010	AAV2-*ND4*	AAV2	CMV	5′UTR COX10	3′UTR COX10(MTS)	*ND4*	bGH	HEK293, HSV1-rc/ΔUL2
LHON	2011	scAAV2-P1*ND4*v2	AAV2-tYF	smCBA		ATP1(MTS); WPRE	*ND4*	bGH	HEK293
LHON	2014	GSO10	AAV2	CMV	β-globin	COX10(MTS)	*ND4*	3′COX10	HEK293
Stargardt	2011	SAR422459	EIAV	CMV			*ABCA4*	SIN-LTR	HEK293
CHM	2011	AAV2.*REP1*	AAV2	CAG	β-globin^SD/SA^	WPRE	*CHM*	bGH	HEK293
CHM	2015	AAV2.*REP1*	AAV2	CAG	β-globin		*CHM*	bGH	HEK293
RP	2011	AAV2.*MERTK*	AAV2	hVMD2	SV40^SD/SA^		*MERTK*	SV40. bGH	HEK293
Usher	2012	UshStat	EIAV	CMV		WPRE	*MYO7A*	SIN-LTR	HEK293
Usher	2018	QR-421a				AON-USH2A			Synthetic
LCA	2019	EDIT-101	AAV5	U6; hGRK1	SV40^SD/SA^	gRNA-CEP290	*SaCas9*	Synthetic	HEK293
LCA	2019	AAV5.*GUCY2D*	AAV5	hGRK1	SV40^SD/SA^		*GUCY2D*	bGH	HeLaS3
XLR	2015	AAV2-tYF.*RS1*	AAV2-tYF	smCB	β-globin^SD/SA^	WPRE	*RS1*	SV40	rHSV/sBHK
XLR	2017	scAAV8-*RS1*	AAV8	hRS1	RS1	IRBP enhancer	*RS1*	Human β-globin	HEK293
ACHM	2015	AAV2-tYF.*CNGB3*	AAV2-tYF	PR1.7	SV40^SD/SA^		*CNGB3*	SV40	rHSV/sBHK
ACHM	2015	AAV.*CNGA3*	AAV8	hCAR		WPREm	*CNGA3*	bGH	HEK293
ACHM	2016	AAV8.*CNGA3*	AAV8	hG1.7			*CNGA3*	SV40	HEK293
ACHM	2016	AAV8.*CNGB3*	AAV8	hCAR			*CNGB3*	SV40	HEK293
ACHM	2019	AGTC-402	AAV2-tYF	PR1.7	SV40^SD/SA^		*CNGA3*	SV40	rHSV/sBHK
RP	2017	AAV8.*RPGR*	AAV8	hGRK1			*RPGRco-ORF15*	bGH	HEK293
RP	2017	AAV-*RPGR*	AAV5	hGRK1	SV40^SD/SA^		*RPGRco-ORF15-Long*	SV40	HEK293
RP	2017	AGTC-501	AAV2-tYF	hGRK1	SV40^SD/SA^		*RPGRco-ORF15*	SV40	rHSV/sBHK
RP	2017	RST-001	AAV2	CAG	β-globin^SD/SA^	WPRE	*Chop2/ChR2*	bGH	HEK293
RP	2017	GS030	AAV2-7m8	CAG			*ChrimsonR-tdT*	bGH	?
RP	2020	BSO1	AAV?	?			*Chr90-FP*	?	?
RP	2017	AAV5.*PDE6B*	AAV5	hGRK1			*PED6B*	bGH	HEK293
RP	2017	CPK850	scAAV8	sRLBP1	mSV40^SD/SA^		*RLBP1*	SV40	HEK293

Ordered on registration date (year) and disease. Full description, size (bp) of elements, and citations can be found in Appendix A.

**Table 2 ijms-21-04197-t002:** Common ubiquitous promoters for rAAV-based ocular gene therapies.

Ubiquitous Promoters	Size (bp)	Origin, Cell Expression, Strength	References
CAGGS aka CBA or CAG	1600	Ubiquitous, +++. Cytomegalovirus immediate-early enhancer, chicken β-actin promoter, chimera between introns from chicken β-actin and rabbit β-globin. pDRIVE CAG plasmid (Invivogen, San Diego, Calif.; having 100% sequence homology with the pCAGGS). The University of Pennsylvania considers CBA and CAGGS the same.	[63]
mini CAG (SV40 Intron)	800	Ubiquitous, +++	[64]
Mini CAG no intron	250	chicken β-actin promoter, Ubiquitous, +	[57]
CBA/CB7	800	Ubiquitous, ++	[65]
smCBA	953	Ubiquitous, ?	[66]
CBh	800	CBA.MVM Ubiquitous, ++	[54,67]
MeCP2	229	ubiquitous	[68]
CMV	800	Ubiquitous, ++, prone to silencing	[54]
shCMV	220	Ubiquitous, ++	[24]
CMVd2	52	Low basal activity. Ubiquitous, Promega, +	[69] Cat.: pFN23A Halo Tag CMV d2
core CMV	30	Not active without enhancers	[48]
SV40mini	106	SV40 minimal promoter	[48,49]
SCP3	81	Super core promoter. (TATA box, Inr, MTE and DPE)	[48]
EF1-α	2500	Ubiquitous, ++	[51,70]
PGK	426	Ubiquitous, ++	[53]
UbC	403	Ubiquitous, ++	[70]

The relative strength (+ being the weakest and +++ being the strongest). Adapted from [57].

**Table 3 ijms-21-04197-t003:** Retina cell-specific promoters in rAAVs for ocular gene therapy.

Müller Glial Cells	Size (bp)	Origin, Cell Expression, Strength	References
CHX10	164	Retinal progenitor cells	[98]
GFAP	2600	Müller glial cells,	[99,100]
GFAP	2200	Müller glial cells (Novartis)	[101]
GfaABC1D	686	Müller glial cells	[96,97]
HRSE-6xHRE-GfaABC1D	~820	Hypoxia-induced reactive MGC promoter. HRE is (A/G)CGT(G/C)C. HRSE from metallothionein II promoter (90 bps)	[97,102]
RLBP1	2789	Müller glial cells	[24,89]
Short RLBP1	581	Müller glial cells	[101]
Murine CD44	1775	Müller glial cells	[24,95]
Murine shCD44	363	Müller glial cells	[24,103]
ProB2	592	Müller glial cells	[50]
**Photoreceptor Cells**	**Size (bp)**	**Origin, Cell Expression, Strength**	**References**
Mouse RHO	1400	Rod PRCs	[104]
Human RHO (rhodopsin)	800	Rod PRCs	[105]
Human RHO	520	Rod- PRCs	[24]
Mouse rod opsin mOp500	500	Rod-PRCs −385/+86	[106]
Mouse rod opsin	221	Rod-PRCs	[107]
Human Rhodopsin kinase (RHOK/GRK1)	294	Rod and cone PRCs. AY327580.1: bp 1793–2087 (−112 to +180). More efficient than IRBP in NHP for cone transduction	[24,91,108,109,110]
Human blue opsin HB570	570	S-cone and subset of M-cones PRCs	[111]
Human blue opsin HB569	569	blue cone opsin PRCs	[106,112]
PR0.5	496	Red cone PRCs	[106]
PR1.7	1700	Red cone PRCs	[106]
PR2.1	2100	Red cone PRCs	[106]
3LCR-PR0.5	~600	Red cone PRCs	[106]
Mouse blue opsin (mBP500)	500	Mouse S opsin	[113]
Human interphotoreceptor retinoid binding protein (hIRBP)	235	Cone and rod PRCs X53044.1, bp 2603–2837	[114]
IRBPe/GNAT2	500	Cone PRCs	[115]
Mouse CAR/ARR3	500	Cone PRCs, some rods, and RPE	[115]
Human CAR/ARR3	405–500	Cone PRCs, some rods, and RPE cells	[115,116]
CAR/ARR3	215	Cone PRC	[117]
Human red opsin	2100	Human red cone opsin	[118]
Human green red opsin (G1.7p)	1700	Cone PRCs. Core green opsin promoter including a mutation (0.5 kb) + Locus Control Region (LCR; 1.2 kb) upstream of the red opsin gene	[119,120,121]
Crx2kb	2000	Cone and rod PRCs	[122]
ProA1	2000	cone PRCs	[50]
ProA4	2000	cone PRCs	[50]
ProC1	731	Cone and rod PRCs	[50]
ProA6,ProB5,ProC22,ProC32,ProD2,ProD3, ProD4,ProD5,ProD6	1229, 619 774, 814, 366, 691, 552, 321, 448	rod PRCs	[50]
Synp161	150	Mouse CD47 enhancer + SV40-mini promoter. Rod PRCs	[49]
**Bipolar Cells**	**Size (bp)**	**Origin, Cell Expression, Strength**	**References**
Mouse metabotropic glutamate receptor 6 (mGrm6)	200	On-bipolar cells	[98]
4× mGRM6e+SV40	1000	On-bipolar cells. 203 bp SV40 minimal promoter	[123]
Grm6e-Chx10-Cabp5	809	200 bp Grm6 + 164 bps Chx10 enhancer + 445 bp Cabp5 promoter. Wide overlapping bipolar expression	[98]
Grm6-SV40	400	Grm6=mGluR6. 200 bp mGluR6 enhancer + SV40 promoter. On-Bipolar cells	[98]
Cabp5	445	Bipolar cells	[98]
Chx10-SV40	364	164 bp Chx10 enhancer + 200 bp SV40 promoter. Bipolar cells and Müller glial cells	[98]
Grm6-mGluR500P	700	On-bipolar cells.	[124]
In4s-In3e- Grm6-mGluR500P	1997	690 bp shortened Intron 4s + 807 bp Intron 3 + 500 bp mGluR500P	[124]
ProB4	1317	Off-bipolar cells	[50]
**Amacrine Cells**	**Size (bp)**	**Origin, Cell Expression, Strength**	**References**
ProC2	964	All amacrine cells + few MGCs	[50]
ProB1	394	Amacrines with processes in one stratum	[50]
**Horizontal Cells**	**Size (bp)**	**Origin, Cell expression, Strength**	**References**
ProC3	694	Some off-target in amacrine and ganglion cells	[50]
**Retinal Ganglion Cells**	**Size (bp)**	**Origin, Cell expression, Strength**	**References**
Syn1	495	Off target amacrine, strength: ++	[125]
Nefh	2251	Strength: +++	[90]
hSNCGp	948	Human SNCG promoter (−785 to +163 region)	[126]
ProA3	2000	Synthetic	[50]
Ple344	801	Gene TUBB3. GCL and corneal nerves. ++	[127]
Ple345	2693	Gene NEFL. +++ (stronger than smCBA)	[127]
**RPE**	**Size (bp)**	**Origin, Cell Expression, Strength**	**References**
hRPE65p	1383	Chr1.68449936-68451318. RPE+ some PRC infection	[128]
NA65p	1383	Codon optimized hRPE65p+SV40 intron + Kozak seq, 150× more efficient than CBA and 300× more efficient than hRPE65p	[36]
VMD2	646	NG_009033.1, bp 4870–5516	[126,129]
Synpiii	1317	+ SV40 mini promoter	[130]

The relative strength (+ being the weakest and +++ being the strongest). Adapted from [57].

**Table 4 ijms-21-04197-t004:** Other elements in rAAV vectors: A. Introns, PRE, and enhancers. B. Miscellaneous.

Introns and PRE and Enhancers	Size	Description, Strength	References
CE (CMV early enhancer)	431	+++, 1.5–67× increase; −118/−522 TSS pCMVβ/5′CMV enhancer	[149]
IRBPe	235	human interphotoreceptor retinoid-binding protein proximal enhancer. Upstream nt −1619/−141 *IRBP*	[115]
metabotropic glutamate receptor 6 enhancer (Grm6e)	200	Grm6 proximal enhancer	[98]
Woodchuck Hepatitis Virus PRE (WPRE)	600	+++, 6–10× increase	[139,150]
Hepatitis B Virus PRE (HPRE)	533	+++, 6–10× increase	[150]
WPRE3	247	++, 6× increase	[139]
MVM	67–97	+++, minute virus of mice, 10× increase	[143]
chCMV.HBB2	~506	Chimeric CMV (146 bp) + human β-globulin intron 2 (340 bp) + exon 3 20 bp incl SA/SD	[151]
Hybrid adenovirus SD^#^_/_IgG Sa*	230	+++, pAdβ, 2× increase to synthetic polyA	[149]
SV40 late SD^#^_/_Sa* (19S/16S)	180	+, pCMVβ (Promega; 1.6× increase)	[149]
Modified SV40 SD^#^_/_Sa*	157	modSV40 SA/SD= modified SV40 splice acceptor/donor intron, 157 bp in length, nucleotides 502–561 and 1410–1497 of SV40 genomic sequence (NC_001669.1) + connecting sequence CGGATCCGG between two fragments.	[101,152]
Mini SV40 SD^#^_/_Sa*	100	Mini SV40 SD^#^_/_Sa* intron	[43,153,154]
Human β -globin intron 2 SD^#^_/_Sa*	875	0.5–86-fold increase. pZac2.1	[139,155,156,157]
F.IX truncated intron1	300	+, human factor IX (100×)	[143,158]
**Miscellaneous**	**Size**	**Description**	**References**
2A	75	Self-cleaving linker	[159]
internal ribosomal entry site (IRES)	600	Ubiquitous. Placed between two genes. The second gene is transcribed without a promoter (at a lower expression compared to the first gene)	[160]
SPTP	154	Synthetic polyA signal/transcriptional pause site frp, pGL4.25	[161]
PolII miR-155	~500	Block-iT PolII miR vector system based on miR-155 expressing artificial miRNAs engineered to a target sequence resulting in target cleavage	[162]
shRNA-YB1	N/A	7-to-45 fold AAV production increase in physical titer	[163]
MIP backbone	N/A	mini-intronic plasmid (MIP) backbones for AAV production increased transgene expression by 40–100 fold in vivo	[144]
R6K	545	+ (~40×),pUC + prokaryotic RNA-OUT antibiotic-free, minicircle AAVs	[144]
OIPR	1300	+ (~40×),pUC + prokaryotic RNA-OUT antibiotic-free, minicircle AAVs	[144]
Shorter OIPR	500	+ (~5×),pUC + prokaryotic RNA-OUT antibiotic-free, minicircle AAVs	[144]

The relative strength (+ being the weakest and +++ being the strongest). Adapted from [7,57,164].

**Table 5 ijms-21-04197-t005:** Polyadenylation sequences.

Polyadenylation	Size	Description, Strength	References
SV40 late	135	+++	[139]
2× SV40 late	100	++/+++	[169]
bGHpolyA	250	++	[149]
2× sNRP1	34	+/++	[167]
Rabbit gbpA	56	Rabbit β-globin	[149]
spA	49	+/++ (7× lower than bGHpolyA, 3× lower than SV40 late)	[139,149]
hGHpolyA	624	+	[41,170,171]
1× sNRP1	17	+	[167]
HSV TK poly(A)	48	herpes simplex virus (HSV) thymidine kinase (TK) polyadenylation signal. Generally used for NeoR and KanR genes	[172]
Adenovirus (L3) USE	21	+	[169]

The relative strength (+ being the weakest and +++ being the strongest). Modified from [7,57,164].

**Table 6 ijms-21-04197-t006:** Inducible promoters.

Inducible Promoters	Size (bp)	Origin, Cell Expression, Strength	References
MT-1	13,200	Zinc, cadmium or copper-inducible sheep metallothionine-Ia promoter	[174]
MMTV LTR	792	dexamethasone (Dex)-inducible mouse mammary tumor virus. Active when glucocorticoids or progestins present	[175,176]
Ptet	270	tetracycline On or Off system promoters (Ptet). 6× mutated TRE (~200 bp) core CMV (~40 bps)	[177]
T7lac	42	T7 bacteriophage promoter (17 bp) requires T7 RNA polymerase and lac operator (25 bp). Induces expression by IPTG	[180]
Riboswitches	~100	ligand-sensing aptamer, a communication module (linker), and an effector domain (ribozyme)	[173,178,179]

Adapted from [57].

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
