# Peer review of "Recombinant Adeno-Associated Viral Vectors (rAAV)-Vector Elements in Ocular Gene Therapy Clinical Trials and Transgene Expression and Bioactivity Assays"

_ijms, 2020, doi:10.3390/ijms21124197_

Round 1

Reviewer 1 Report

The ms. is deemed acceptable to be published in its present form.

Author Response

No further comments by Reviewer 1.

Reviewer 2 Report

The authors have compiled a comprehensively detailed manuscript about the use of rAAV vectors and promoters used in retinal gene therapy. I have the following suggestions:

  1. The authors should also discuss a little more the potential use of AAVs in treating anterior chamber diseases, such as glaucoma.
  2. In the mini gene section, a quick literature search showed a publication on minicep290 that showed transient efficacy in a mouse mutant. The authors should discuss this strategy. This is also an upcoming field based on the information presented at ASGCT 2020. Other genes such as ABCA4 and USH2A are also being considered as a minigene approach.

Author Response

Response: We adjusted the section 4 to accommodate the changes (see below)

Subtitle (line 571):

“4.1. Intron removal, exon removal, surrogates, and pathway-modifying therapies”

Minigenes (line 588)

Shortened versions of proteins (exon truncations aka minigenes)

Successful minigene supplementation therapies (Line 598-607)

“Also, the CEP290 gene (7.5 kb cDNA coding for 2,479 amino acids) is too large to fit in an rAAV vector. A smaller version of the CEP290 cDNA (minigene miniCEP290580–1180 coding for the RAB8A-binding domain) temporarily rescued in part the ciliary length and retinal function in the Cep290rd16 mouse [187]. The company Ophthotech (now Iveric) further explores the miniCEP290580–1180, a miniABCA4 (no published information available), and miniUSH2A (no published information available) for rAAV vector therapy. The USH2A gene (15,6 kb cDNA coding for 5202 amino acids) cDNA has been shortened to the miniUSH2A-1 (~6,8 kb) and miniUSH2A-2 (~4,1 kb) and delivered by Tol2 transposase mRNA into homozygous one-cell staged ush2armc1 zebrafish embryos restoring visual motor responses and retinal function [188].”

+

Antiangiogenic factors (line 619-646) + fluorescent proteins (line 647-661)

“The identification of key players in signaling pathways (e.g. VEGF, TGF-beta, Wnt) is an exciting but complex research area. Here, exemplified on glaucoma therapies, the expression of a key player is altered (e.g. by antibody or viral vector administration) to induce a lasting change on intraocular pressure (IOP). The search for key players is generally on: (a) mutated genes that are more frequently found in patients with glaucoma (single nucleotide polymorphisms [SNP] databases; e.g., CAV-1), (b) genes affected by glaucoma (e.g. data from [single-cell] RNAseq healthy-disease condition databases; e.g., ANGPTL7, MMP1, and PLAT), or (c) gene products that are actively involved in regulating the intraocular pressure (e.g. data from knock-out/in gene library screens and literature searches; e.g., RhoA-Rho kinases and prostaglandin EP4 agonists) [211]. Similar approaches are explored for the identification of pathway genes for RP and LCA.

The expression of pathway-modifying factors can be done by the injection of neuronal progenitor cells that release growth factors (jCyte, Inc, product jCell; ReNeuron Limited, product hRPCRP) or viral vectors expressing the specific key player (see Figure 1, Table 1 rAAV products for AMD / glaucoma). Cell survival factors, such as pigment-endothelial-growth factor (PEDF), are discussed in the section 3.3. For example, the suppression of the vascular endothelial growth factor receptor (VEGFR) activation and the complement cascade activation in the RPE and choroid by repeated intravitreal antibody injections is successfully used in clinical studies [204]. This led to the development of rAAV gene cassettes that express upon infection a soluble protein fragment that partially binds to the VEGF receptor or complement system proteins inhibiting the disease pathway cascade. The vectors have efficiently prevented a full-blown AMD phenotype in AMD models and are currently tested in clinical trials: (1) the small soluble fms-like tyrosine kinase-1 (sFlt-1. Non-membrane associated splice variant of VEGFR1 encoded by the FLT1 gene; Adverum Biotechnologies; Sanofi Genzyme) [41,205,206], (2) endostatin (cleavage product of collagen XVIII) / angiostatin (cleavage product of fibrinogen; Oxford Biomedica) [207], (3) the complement factor I (CFI, a C3b/C4b inactivator in the complement cascade; Gyroscope Therapeutics) [208], (4) anti-VEGFfab Heavy chain & anti-VEGFfab Light chain (Regenxbio) [209], and (5) the soluble CD59 antigen (binding to C5b preventing C9 incorporation in the complement cascade; Hemera Biosciences) [210].

Finally, the expression of light-activated opsin-like proteins are explored to restore (partial) vision in RP patients (see Table 1 rAAV products RST-001, GS030, and BSO1). Three different fluorescent proteins (ChR2 aka Channelrhodopsin-2 [930 bp], ChR88m19-tdTomato aka ChrimsonR [2496 bp], and Chr90 aka Chronos [1710 bp]) are explored in clinical trials [212,213]. Chronos-GFP (green-shifted) and ChrimsonR-tdTomato (red-shifted) are second generation fluorescent proteins that have faster kinetics and are more light sensitive compared to ChR2. Further tests are needed to find out which approach restores optimal vision in patients with ocular diseases: (1) a wildtype or surrogate gene (e.g. CRB1 or CRB2) to a CRB1-RP patient, (2) a shortened gene supplementation (e.g. miniCEP290580–1180), (3) neuroprotective factors (e.g. CNTF), (4) the delivery of antiangiogenic factors (e.g. sFLT1), or (5) disease pathway-modulation (e.g. ANGPTL7 in glaucoma), or (6) the introduction of a light-activated protein. We will mainly focus on the supplementation of a wildtype copy of a gene for the rest of the review.“

+

’Concluding remarks’ under bullet point 6 (line 1261)

“(6)        Surrogate gene (homolog/ortholog or synthetic) and minigene supplementation may circumvent cellular immunogenicity (Section 4.1).”

Reviewer 3 Report

What is required of a good review article is

1) whether the articles referenced are properly selected, and 2) whether the review article summarizes what we want to know. Although I concerned the below mentioned issues, it is worth posting as a review of gene therapy in the IJMS.

1) 320 papers referenced may be too much. It would be better to select adequately.

2) It is hard to understand which vector and which method is better in this review. However, endpoints vary depending on clinical trials in gene therapy, and meta-analysis may be difficult. Therefore, it is tough for authors to request it.

Author Response

Response: We completely agree that 320 papers referenced are much more than provided in usual reviews but this a meta-analysis of elements used in ocular rAAV vectors that present usual lists to interested readers. The large amount of references relates to the very large research field covered in the review (clinical trials, gene cassette vector elements, AAV capsid tropism, ITRs and vector integration into the genome, screening assays, new techniques). We believe that the (primary data) references are needed for the readers so they can quickly find the correct publication related to the statement.

Reviewer 4 Report

Thank you for letting me revise this very interesting manuscript. This is a comprehensive literature review that summarizes a huge amount of information about the actual status on retinal gene therapy. The present manuscript is very well written and organized, however my only concern is the size of this work; it is massive. It took me a long time to read it, but I learned many things that I did not know.

There are some format adjustments that need to be done in order to be published in IJMS; highlights and index are not normally required. Also, back matter lacks of authors contributions.

References in the reference list are not uniform, some references use the abbreviated page range (eg. 371-72) and some others use it the entire number (eg. 371-372). Some other references are lacking page range (eg. Ref. 21).

Author Response

Response: We removed the highlights and the index. We also revised the back matter (author contribution added, financial contribution updated to IJMS format) and references (e.g. 371-72 changed to 371-372). Some journals do not want to indicate page ranges such as for the Ref 21. Pubmed.ncbi indicates for this publication the same citation we use without page range: Duong TT, Lim J, Vasireddy V, et al. Comparative AAV-eGFP Transgene Expression Using Vector Serotypes 1-9, 7m8, and 8b in Human Pluripotent Stem Cells, RPEs, and Human and Rat Cortical Neurons. Stem Cells Int. 2019;2019:7281912. Published 2019 Jan 17. doi:10.1155/2019/7281912

Back matter changes:

Line 1305-1309 Material and methods:

“The data was curated in MS Excel (Microsoft software) and then loaded in GraphPad Prism version 7 (GraphPad software). Figure 1, 2, 4, and 5 were generated in MS PowerPoint (Microsoft software). All figures were exported as vector graphs and imported in Photoshop (Adobe Software) for final type-set editing.”

Line 1311-1313: Statistical analysis

“Statistical analyses and visualization for Figure 3 was done in GraphPad Prism version 7 (GraphPad software).”

Line 1324-1326 Acknowledgments: “This research was funded by Foundation Fighting Blindness [TA-GT-0715-0665-LUMC] and The Netherlands Organization for Health Research and Development [ZonMw grant 43200004].

Line 1328-1332: Author Contributions

Conceptualization, T.M.B. and J.W.; Methodology, T.M.B; Validation, T.M.B, and J.W.; Formal Analysis, T.M.B; Investigation, T.M.B.; Resources, J.W.; Data Curation, T.M.B.; Writing – Original Draft Preparation, T.M.B.; Writing – Review & Editing, T.M.B. and J.W.; Visualization, T.M.B.; Supervision, J.W.; Project Administration, J.W.; Funding Acquisition, J.W.

Line 1334-1336: Supplementary Materials

Table S1: Description of clinical trial rAAV gene therapy products in ophthalmology / Proviral plasmids. Includes the raw data for the generation of Table 1-5, Figure 2 and Figure 3.”

This manuscript is a resubmission of an earlier submission. The following is a list of the peer review reports and author responses from that submission.

Round 1

Reviewer 1 Report

The authors have complied with this reviewer's queries in a satisfactory fashion. The ms. is deemed publishable in its present form.

Reviewer 2 Report

I genuinely like the concept of carrying out a meta-analysis of all the current gene therapy studies, but overall the text remains poor. Despite some revisions most sections still essentially consist of little more than lists of extremely short statements that have little to no interconnectivity, making it extremely tiresome to follow the point the authors are trying to get across. An excellent example would be lines 181-183, a passage of text containing three sentences in just three lines that would be much easier to read if it were one connected sentence that flowed. The lines that follow (184-190) contain another seven sentences in seven lines, including one only 4 words in length. The review contains good information, but the never-ending stop-start sentence structure and lack of overall cohesion makes it almost painfully difficult to read or understand.

In keeping with the text mostly reading like a list of statements, there is also a distinct lack of interpretation throughout. One example (of many, many) would be lines 449-451, where they mention the NA65p promoter having 150x/300x ‘higher potency’ than the hRPE65 promoter, but provide no information to the reader on how the promoter was modified, or whether such high levels of expression are desirable, or whether it is now less likely to be down-regulated in a disease state?

Another example is lines 480-481 that says in its entirety ‘The tissue-specific enhancers increased expression but were less tissue specific’. How? Why? Is this good or bad? When might it be worth trading specificity for higher expression? As with so many statements in this manuscript, it is just dumped out there with no interpretation to help the reader.

As the articles stated intention is to ‘provide an intuitive workflow to design novel vectors’ it does a very poor job of helping the reader make informed choices as to say, which promoter might be most appropriate (ubiquitous, cell-specific, strong, weak etc.), as in most examples the article doesn’t give any commentary on the pros and cons of various elements.

In summary, I cannot recommend strongly enough that the text be submitted to a scientific copy editor to address the manifold issues throughout and improve readability.

While the authors have made many changes to the text it is still littered with grammatical and typographical errors, summarized (not exhaustively) below:

Line 101: Change ‘focused’ to ‘focusing’

Line 106: Change ‘ex vitro’ to ‘ex vivo’

Line 125: Remove extra full stop/period after ‘disease.’

Line 125: Change ‘preventing’ to ‘preventative’

Lines 129-130: Badly phrased and should be reworded.

Figure 1: The authors have not included intracameral injection as a delivery route

Line 156: Remove extra comma after ‘i.m.,’

Line 157: Change ‘intravitreous’ to ‘intravitreal’

Line 180: ‘disorders of the visual function’ does not make sense and needs to be rephrased.

Line 203: Why is ‘nuclear localization’ italicized?

Line 206: The sentence ‘What cells are infected?’ is included without any context and should be removed.

Figure 2: The authors should make it clear why they included the genes they have, as this is quite an abridged list. Most prevalent (above X% incidence) perhaps?

Line 257: ‘we observed yet’ does not make sense and needs to be rephrased.

Line 327: Remove ‘only’

Line 329: Remove the extra full stop/period after [81 bp]

Line 331-333: The last clause of this sentence doesn’t make sense and should be rephrased.

Line 337: Add a comma after ‘DNA’

Line 351: ‘Ubiquitous promoters transcribe relatively stable expression’ is not true and is indeed contradicted below when discussing CMV silencing.

Lines 371-373: Is this sentence trying to make a point about serotype selection?

Line 374: To what is ‘The studies’ referring to?

Lines 408-411: This isn’t entirely true. Some dominant diseases do respond positively to gene augmentation.

Lines 418-423: The authors state that expression of both a therapeutic and neuroprotective transgene from a single construct is ‘very promising’, but do not reference any studies that have done so. Please add these to the end of line 423.

Lines 433-434: The authors should acknowledge in this section that the use of native promoters is also problematic, wherein they are likely to be down-regulated in diseased tissues, as with RPE65.

Line 480: Add a comma after ‘expression’

Lines 492-494: And… what were the results? Summarize for the reader.

Lines: 498-499: Explain why they might be redundant when used in combination with introns.

Line 515: Change ‘to suppress’ to ‘suppressed’

Lines: 525-531: What relevance does this have to the design of therapeutic vectors? Presumably as a serotype screening assay, but again, there is no interpretation or explanation.

Line 545:  Add a comma after ‘(919 bp)’

Line 546: Remove ‘But’

Line 547: Remove the comma following ‘(50 bp)’

Lines: 549-551: Combine these sentences

Lines 581-582: This sentence is redundant and should be removed.

Line 584: ‘rendering the plasmid to a Tet-off system’ does not make sense and should be rephrased.

Line 595-596: This information is incorrect. Riboswitches can be incorporated at both the 5’ and 3’ ends of expression constructs, often at the same time.

Line 607: Change ‘gene’ to ‘transgene’

Line 610: Change ‘AAV’ to ‘rAAV’

Line 613: Change ‘on highly’ to ‘of highly’

Line 617: Change ‘encoded on’ to ‘encoded by’

Line 650: Change ‘are in’ to ‘have been in’

Lines 650-652: Something cannot still be ‘in clinical trials’ if the trial has already ended. Please rephrase.  

And on and on and on... there are simply too many mistakes in the manuscript to try and fix them all.